# Implicit Reconstruct Spatiotemporal Super-Resolution Microscopy in Arbitrary Dimension

## Abstract

High-resolution 4D fluorescence microscopy imaging, essential for deciphering dynamic biological processes, is typically challenged by insufficient spatiotemporal resolutions, including restricted t-axis sampling density to prevent photobleaching and issues with anisotropic resolution. To address these challenges, we propose an implicit neural representation-based arbitrary scale super-resolution framework, termed *SpatimeINR*, which leverages spatiotemporal latent representation in conjunction with a multilayer perceptron for 4D rendering, while incorporating cycle-consistency loss to ensure fidelity with the original data. Extensive experiments on lung cancer cell and *C.elegans* cell membrane fluorescence datasets demonstrate that our approach can accurately reconstruct the nonlinear dynamic motion of biological samples along the time axis and significantly outperforms state-of-the-art methods in both temporal and spatial (4D) super-resolution tasks. Code and data will be released after the review process.

## 1 Introduction

In recent years, driven by the increasing demand for detailed characterization of dynamic processes in biological research, 4D imaging via fluorescence microscopy (*i.e.*, three-dimensional space with an added temporal dimension) has gradually become an indispensable tool for decoding cellular behaviors, tissue development, and complex cell interactions (Cao et al., 2020; Guan et al., 2025; Murray et al., 2008; Zhao et al., 2024; Mertz, 2019). However, obtaining high-resolution 4D images faces two major challenges. On one hand, to avoid issues such as phototoxicity and photobleaching caused by overexposure(Tinevez et al., 2012; Hoebe et al., 2007; 2008), traditional imaging systems must acquire data at lower sampling rates, which leads to insufficient resolution along the time axis and hampers the capture of complete spatiotemporal information. On the other hand, the volumetric data obtained by microscopes often exhibit anisotropy, with high resolution in the XY plane but relatively coarse resolution along the Z-direction, which further complicates high-quality reconstruction (Li et al., 2025; Fang et al., 2024; Williams & Drew, 2019). Furthermore, for temporal super-resolution processing, it is not only necessary to enhance the time-domain resolution but also to ensure that the reconstructed dynamic process authentically reflects the actual motion trajectory of the biological samples, thereby accurately capturing their inherent nonlinear and aperiodic dynamics (Appendix C).

Existing approaches (Cao et al., 2018; Sokooti et al., 2017; Yang et al., 2017) for 4D medical image interpolation and super-resolution can be divided into two main categories. The first involves traditional interpolation techniques based on optical flow estimation, such as video frame interpolation (VFI) methods (Chen et al., 2022), which generate intermediate frames by computing the motion field between consecutive frames. The second category employs deep learning models that generate intermediate frames by sampling only the initial and final frames of the medical images (Kim et al., 2024; Guo et al., 2021), achieving satisfactory results under conditions of periodic and relatively smooth motion. However, these methods rely on the input of the starting and ending frames, making them suitable for medical image registration tasks while being less applicable to super-resolution tasks on low-resolution 4D sequences. This limitation hampers the ability to capture continuous dynamic details in biomedical image sequences

characterized by complex and non-periodic motion, ultimately leading to reconstructed outputs that are prone to artifacts or loss of crucial information.

To address the above challenges, Arbitrary Scale Image Super-Resolution (ASISR) has emerged as a novel image reconstruction paradigm that offers the potential to overcome the limitations inherent in fixed-scale super-resolution (Chen et al., 2025; Peng et al., 2025). By constructing a continuous image representation, ASISR enables high-quality reconstructions at arbitrary scales, thereby preserving fine details while mitigating information loss caused by fixed sampling scales (Chen et al., 2021b; Wu et al., 2022; Xie et al., 2025; Chen et al., 2023). To date, most approaches based on ASISR have predominantly focused on conventional two-dimensional images or three-dimensional medical images. Mature solutions for directly achieving spatiotemporal super-resolution on low-resolution continuous 4D imaging data still remain elusive.

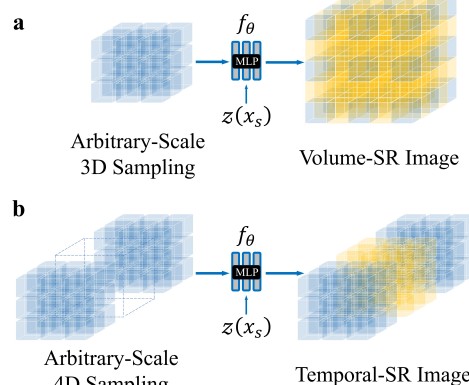

**Figure 1:** (a) and (b) demonstrate that *SpatimeINR*, through dense sampling across arbitrary dimensions, can reconstruct spatiotemporal super-resolution images at arbitrary resolutions.

In this work, we propose an implicit neural representation (INR) based arbitrary scale super-resolution method, with the aim of achieving arbitrary scale reconstruction for 4D data (*i.e.*, three-dimensional spatial data with a temporal component). Specifically, we construct a *SpatimeINR* model that implicitly represents the 4D spatiotemporal data as a continuous parameterized function, maintaining continuity in both spatial and temporal domains. By implicitly mapping discretely sampled and the corresponding spatiotemporal features to high-dimensional vectors, we obtain a continuous representation of the high-resolution volumetric data (Figure 1). Instead of relying only on boundary frames, we locally sample the low-resolution data. The conditional MLP predicts voxel values, with cycle consistency ensuring faithful reconstruction. We evaluate *SpatimeINR* on the lung cancer cell (A549) (Maška et al., 2023; Castilla et al., 2018; Sorokin et al., 2018) and *C.elegans* cell membrane fluorescence (CE) microscopy datasets (Zhao et al., 2024), covering both spatiotemporal and spatial super-resolution tasks. The experimental results demonstrate that *SpatimeINR* significantly outperforms traditional methods and deep learning methods, including PSNR, SSIM, LPIPS, Dice coefficient and Biological Shape Accuracy (BSA). Overall, the framework effectively captures complex spatiotemporal information and achieves high-quality reconstruction for non-periodic and dynamically changing 4D data, providing a robust foundation for the recovery of fine details along all axes.

Our main contributions can be summarized as follows:

1. We propose an INR-based *SpatimeINR* model that effectively addresses the issue of insufficient spatiotemporal resolution in time-lapse microscopy imaging by performing arbitrary-scale super-resolution on complete 4D continuous data across both spatial and temporal dimensions.

2. We design a hybrid framework that combines spatiotemporal latent representations with implicit volumetric rendering, effectively capturing dynamic variations in complex biological samples.

3. Experimental results demonstrate that on the *A549* and *CE* datasets, our method achieves higher PSNR, SSIM, LPIPS, Dice and BSA than existing methods. Ablation studies confirm the vital role of each module, demonstrating the framework's efficacy in mitigating issues such as insufficient sampling and anisotropic resolution.

## 2 RELATED WORK

### 2.1 ARBITRARY SCALE IMAGE SUPER-RESOLUTION

In the field of image reconstruction and super-resolution, image representation methods can be broadly divided into explicit and implicit formulations. **Explicit representation** methods rely on

discrete arrays of pixels or voxels, meshes, or polygons, which directly store the sampled values of the image signal (Merks & Glazier, 2005). Traditional super-resolution approaches based on explicit representations include interpolation methods, sparse coding reconstruction, as well as pre-upsampling (Dong et al., 2014), post-upsampling (Lim et al., 2017), progressive upsampling (Lai et al., 2017), and iterative up/down sampling methods (Haris et al., 2018) based on deep learning. Although these methods are intuitive and straightforward, they are limited by the fixed sampling resolution when attempting to capture continuously varying information, rendering arbitrary scale reconstructions problematic (Xie et al., 2025). In contrast, **implicit representation** methods construct a continuous mapping function that associates any coordinate with its corresponding pixel (or voxel) value, thereby overcoming the limitations imposed by discrete sampling (Sitzmann et al., 2020; Mildenhall et al., 2021) and can also be used for multi-view synthesis tasks (Pumarola et al., 2021; Park et al., 2021b;a),. In particular, implicit neural representation (INR) methods leverage multi-layer perceptrons (MLPs) for high-dimensional continuous modeling of images; this allows sampling and reconstruction at arbitrary spatial scales and magnification factors, effectively mitigating information loss. ASISR is proposed within this context with the goal of forming a continuous image representation that balances global structure and local details to achieve high-quality arbitrary scale reconstruction (Chen et al., 2021b; Xie et al., 2025; Chen et al., 2023). Recent work includes MetaSR (Hu et al., 2019), which adopts a meta-learning approach to predict upsampling filter weights for multi-scale reconstruction; LIIF, which describes images as locally implicit functions to restore high-frequency details; SeCo-INR(Ekanayake et al., 2025) proposed an INR method that can be enhanced using segmentation masks; and Equivariant-ASISR (Xie et al., 2025), which introduces rotational equivariance to enhance consistency across different directions. Additionally, VideoINR (Chen et al., 2022) has demonstrated promising results in video spatiotemporal implicit representation by modeling pixel values continuously in the time domain for high-quality video frame interpolation and reconstruction. Nonetheless, for 4D data with more complex spatiotemporal continuity (*e.g.*, 4D medical images), capturing precise spatial structures alongside temporal dynamics remains a challenging open problem.

## 2.2 4D Medical Image Frame Interpolation

For 4D medical imaging, achieving high spatial and temporal resolution is often hindered by constraints such as radiation dose control, extended reconstruction times, phototoxicity, photodamage, and anisotropic resolution in fluorescence imaging (Cao et al., 2020; Murray et al., 2008; Zhao et al., 2024; Li et al., 2025; Fang et al., 2024). To alleviate these issues, medical image frame interpolation methods have become a recent research (Guo et al., 2021; 2020). Traditional interpolation methods typically rely on optical flow estimation and spatiotemporal motion field modeling. Optical flow-based methods compute the motion field between adjacent frames to generate intermediate frames, while other approaches (*e.g.*, based on B-spline (Eilers & Marx, 1996) free-form deformation) estimate displacement vector fields by optimizing local topological similarity (Balakrishnan et al., 2019; Kim & Ye, 2022). Although these methods perform well in scenarios with linear motion, they tend to fall short when dealing with non-linear, complex motions. In contrast, recent deep learning methods offer new strategies for 4D medical image frame interpolation. For example, SVIN (Guo et al., 2020) employs a 3D convolutional neural network to estimate motion fields in an unsupervised manner and generating highly informative intermediate frames. Meanwhile, UVI-Net (Kim et al., 2024) constructs an unsupervised framework based on periodic and cyclic consistency constraints to achieve continuous interpolation of intermediate frames without relying on true intermediate frame supervision. Wiesner *et al.* (Wiesner et al., 2024) proposed an INR method based on a Signed Distance Function (SDF) to generate novel cell morphology sequences and demonstrated its efficacy in temporal interpolation tasks; however, this method is tailored to specific ternary cell morphology data constructed using SDF, which limits its applicability to conventional imaging data with more complex signals. Flow-INR (Saitta et al., 2024) can construct 4D MRI images of blood flow. Dyn-INR (Feng et al., 2025) achieves spatiotemporal super-resolution in the $(x, y, t)$ space; its capability to model the complete 4D space, encompassing all three spatial dimensions along with the temporal dimension, remains insufficient. Furthermore, the employed frame interpolation approach cannot be strictly equated with image super-resolution. Despite these advances, certain limitations remain. On one hand, for biological imaging over extended periods (*e.g.*, multiple cellular division events) and non-periodic motion, these methods struggle to effectively model long sequences. On the other hand, the generated intermediate frames may contain content errors or inconsistencies that detract from overall reconstruction quality.

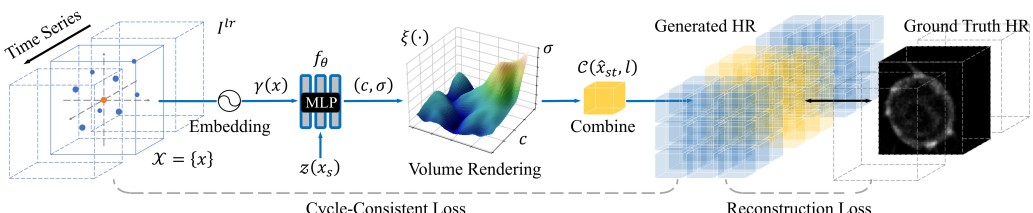

**Figure 2:** An overview of our proposed *SpatimeINR* method.

## 3 METHODS

### 3.1 SPATIMEINR

We propose a method for the spatiotemporal evolution of volumetric images, where the spatiotemporal variations are implicitly represented as a continuous *SpatimeINR* function parameterized by a neural network. Given a low spatiotemporal resolution 4D sequence $I^{lr} \in \mathbb{R}^{h \times w \times l \times t}$, (where the dimensions correspond to three-dimensional space and time), the 4D data is first normalized in each dimension to the range $[-1, 1]$. The training set is then constructed as a sequence of 3D images at $t$ time instants. The spatial domain $(x, y, z) \in [-1, 1]^3$ is discretized into $N$ positions, and the corresponding signal over $t$ time instants is denoted by $\{\tau^i\}_{i=1}^N$, where $\tau^i \in \mathbb{R}^T$ represents the temporal signal at the $i$-th spatial location. Hence, for each 4D image sequence, the ideal continuous representation is obtained from a discrete spatiotemporal sampling set $X = \{u_{st}\}$, with corresponding coordinates $u_{st} = (x, y, z, t) \in [-1, 1]^4$. Thus, the high-resolution volumetric data $I^{hr} \in \mathbb{R}^{H \times W \times L \times T}$ is given by:

$$I^{hr} = \{SpatimeINR_{(\tau^i)}(u_{st})\}_{i=1}^N, \tag{1}$$

In accordance with current advances in ASISR (Chen et al., 2021b; Xie et al., 2025), we propose to approximate the above *SpatimeINR* using a latent variable-encoded conditional multi-layer perceptron. An encoder $\mathcal{E}_\phi$ extracts latent codes from the low-resolution 4D image $I^{lr}$. For a spatial coordinate $x_s = (x, y, z) \in [-1, 1]^3$, the associated latent code is denoted by $z(x_s)$. The multi-layer perceptron $f_\theta$ takes as input the spatiotemporal coordinate $u_{st}$ together with the latent code $z(x_s)$ and outputs the predicted voxel intensity via a volumetric rendering function $\xi(\cdot)$ (Figure 2). For any spatiotemporal query coordinate $u_{st} \in [-1, 1]^4$, we require that

$$\xi\big[f_\theta(u_{st}, z(x_s))\big] \rightarrow SpatimeINR_{(\tau^i)}(u_{st}), \tag{2}$$

so that high-resolution data $I^{hr}$, as well as the super-resolved reconstruction result at any time frame, is obtained from denser spatiotemporal sampling.

### 3.2 SPATIOTEMPORAL LATENT REPRESENTATION

For 4D continuous imaging data, a three-dimensional representation is utilized to aggregate the spatiotemporal correlations and features at each spatial location, *i.e.*, each spatial voxel $(x_i, y_i, z_i)$ encodes all voxel intensities along the temporal axis (Figure 3). Specifically, for each spatial voxel $(x_i, y_i, z_i)$, we stack the $t$ voxel intensities along time into a vector of length $t$:

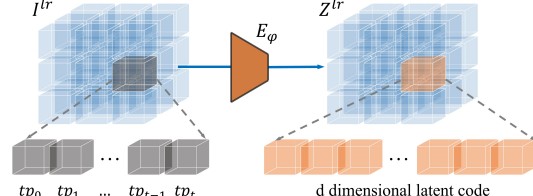

**Figure 3:** The workflow of spatiotemporal encoder.

$$\tau^i = \Big(I^{lr}(x_i, y_i, z_i, t_1),\, I^{lr}(x_i, y_i, z_i, t_2),\, \ldots,\, I^{lr}(x_i, y_i, z_i, t_t)\Big) \in \mathbb{R}^t, \tag{3}$$

Thus, each spatial position is represented by a $t$-dimensional feature vector capturing its full temporal trajectory. In this context, an encoder $\mathcal{E}_\phi$ is employed to extract semantic features from the low-resolution volumetric image $I^{lr} \in \mathbb{R}^{h \times w \times l \times t}$. The encoder $\mathcal{E}_\phi$ takes $I^{lr}$ as input and produces

a feature map $Z^{lr} \in \mathbb{R}^{h \times w \times l \times d}$, where each voxel is transformed into a latent code of dimension $|d|$, capturing both the three-dimensional structure and temporal dynamics:

$$\mathcal{E}_\phi : \mathbb{R}^{h \times w \times l \times t} \to \mathbb{R}^{h \times w \times l \times d}, \quad Z^{lr} = \mathcal{E}_\phi(I^{lr}), \tag{4}$$

For any three-dimensional spatial coordinate $x_s = (x, y, z) \in [-1, 1]^3$, the corresponding latent code is obtained from $Z^{lr}$ via an $L_1$ nearest neighbor lookup, *i.e.*, $z(x_s) = Z^{lr}(x_s)$, where $\phi$ denotes the trainable parameters of the encoder. This process produces a $d$-dimensional latent vector that integrates the spatial features at $(x, y, z)$ in the four-dimensional data $(x, y, z, t)$ with the temporal characteristics from time $0$ to $t$. The query coordinate is subsequently concatenated with this $d$-dimensional vector and fed into an MLP. The encoder is based on and improved upon the residual convolutional neural network proposed in RCAN (Chen et al., 2021a). This per-voxel latent code extraction strategy is critical for enabling the decoder to effectively integrate local image intensity information and recover fine details in high-resolution images, especially at large upsampling scales.

### 3.3 IMPLICIT 4D RENDERING

The implicit neural representation (INR) of medical volumetric data requires continuous-domain rendering to avoid artifacts such as holes, checkerboard patterns, and aliasing that may arise when predicting voxel values independently. Research in the NeRF domain (Mildenhall et al., 2021; Chen et al., 2023) provides a stable, continuous, and artifact-robust volumetric rendering strategy. In this work, we adopt a 4D sampling strategy combined with volumetric rendering to predict the final voxel intensities. Let the target point be $\hat{x}_{st} = (\hat{x}, \hat{y}, \hat{z}, \hat{t}) \in [-1, 1]^4$, Centered at this target point, we construct a 4D hypercube with side length $l$ (where $l$ denotes the distance between adjacent coordinates along each dimension), defined as

$$B_4\left(\hat{x}_{st}, \frac{l}{2}\right) = \left\{ \mu \in \mathbb{R}^4 \,\middle|\, \|\mu - \hat{\mu}_{st}\|_1 \le \frac{l}{2} \right\}, \tag{5}$$

From the uniformly distributed points $\mu$ in this region, we sample $M$ points (in practice, $M = n^4$ with $n \in \{1, 2, 3, \ldots\}$), forming a set $\mathcal{M} = \{x_{st}\}$. To enhance the representation of high-frequency information in the 4D space, a positional encoding function $\gamma(\cdot)$ is applied to each sampled point $x_{st} = (x, y, z, t) \in \mathcal{M}$, mapping it onto a higher-dimensional feature space:

$$\gamma(x_{st}) = \left( \sin(2^0 \pi x_{st}), \cos(2^0 \pi x_{st}), \ldots, \sin(2^{L-1} \pi x_{st}), \cos(2^{L-1} \pi x_{st}) \right), \tag{6}$$

with the hyperparameter $L$ set to 10 by default. Subsequently, the conditional MLP $f_\theta$ is leveraged to predict the voxel intensity function $c(\cdot)$ and the density function $\sigma(\cdot)$. For each encoded sampled point $\gamma(x_{st})$, we have:

$$(c, \sigma) = f_\theta\big(\gamma(x_{st}), z(x_s)\big), \tag{7}$$

within the hypercube $B(\hat{x}_{st}, l)$, it is assumed that the density $\sigma$ at any point $\hat{x}$ depends solely on the L2 distance $r = \|\hat{x} - \hat{x}_{st}\|_2$ from the (Krähenbühl & Koltun, 2011). Accordingly, the density function $\xi(\cdot)$ is defined as:

$$\xi(r, c) = 2\pi^2 \sum_{i=1}^M \frac{r_i^3 \left( 1 - \exp\left(-\sigma_i(r_{i+1} - r_i)\right) \right)}{\exp\left(2\pi^2 \sum_{j=1}^i r_j^3 \sigma_j (r_{j+1} - r_j)\right)} c_i, \tag{8}$$

with the detailed derivation of $\xi(\cdot)$ provided in the Appendix J. After processing the set $\{c_k, \sigma_k\}_{k=1}^M$ via $\xi(\cdot)$, the resulting voxel intensity $C(\hat{x}_{st}, l)$ is given by:

$$C(\hat{x}_{st}, l) = \xi\left(\{c_k, \sigma_k\}_{k=1}^M\right), \tag{9}$$

### 3.4 NETWORK OPTIMIZATION

During training, in order to ensure that the parameterized continuous function closely approximates the defined *SpatimeINR* representation, we use the mean squared error (MSE) as the reconstruction loss. Specifically, for each sampled point $x$, the reconstruction loss is defined as:

$$\mathcal{L}_{rec}\left(\xi\big[f_\theta(u_{st}, z(x_s)\big], \textit{SpatimeINR}_{(\tau^i)}(u_{st})\right) = \left(\xi\big[f_\theta(u_{st}, z(x_s)\big] - \textit{SpatimeINR}_{(\tau^i)}(u_{st})\right)^2, \tag{10}$$

Given the high-resolution output from the INR, we use a down-sampling operator $\mathcal{D}(\cdot)$ to project it back into the low-resolution space. To avoid distortions in the super-resolved result, we compute a cycle consistency loss defined as:

$$\mathcal{L}_{\text{cycle}} = \left\| \mathcal{D}\Big(\xi\big[f_\theta(\{u_{st}\}, z(\{x_s\}))\big]\Big) - I^{lr} \right\|^2, \tag{11}$$

The downsampling operator $D(\cdot)$ is defined via mean downsampling:

$$D(I^{\text{lr}})(x, y, z, t) = \frac{1}{|W|} \sum_{i \in W} I^{\text{lr}}(i), \tag{12}$$

where $|W|$ denotes the number of pixels (or voxels) in the region $W$. In the absence of coarse-scale ground truth, this operator serves as an implicit constraint that alleviates artifacts—such as holes, speckle noise, and local brightness irregularities—arising from direct fitting. Letting the sampling distribution be denoted by $P$, the global objective function is formulated as:

$$\mathcal{L}\big(\theta, \{z_j\}_{j=1}^N\big) = \mathbb{E}_{u_{st} \sim P} \left[ \sum_{i=1}^{N} \mathcal{L}_{\text{rec}}\Big(\xi\big[f_\theta(u_{st}, z(x_s))\big], \textit{SpatimeINR}_{(\tau^i)}(u_{st})\Big) + \mathcal{L}_{\text{cycle}} \right], \tag{13}$$

here, $z_i = z(x_s)$ represents the latent code corresponding to the spatial position in the low-resolution image $I^{lr}$. Using gradient descent, we jointly optimize the parameters $\theta$ of the MLP and the latent codes $\{z_j\}_{j=1}^N$, so that the parameterized continuous function approximator satisfies

$$\textit{SpatimeINR}_{(\tau^i)}(u_{st}) \approx \xi\big[f_\theta(u_{st}, z(x_s)\big], \tag{14}$$

for any spatiotemporal coordinate $x$ and its corresponding latent code $z(x_s)$. This objective ensures that high-resolution 4D images can be reconstructed in a high-quality manner at arbitrary scales, thereby significantly enhancing the model's ability to represent and restore complex spatiotemporal structures.

## 4 EXPERIMENTS

### 4.1 DATASETS

Our experiments utilize two datasets. The first dataset is the A549 human lung cancer cell dataset from the Cell Tracking Challenge (Maška et al., 2023; Castilla et al., 2018; Sorokin et al., 2018). This dataset comprises four time series acquired by a PerkinElmer UltraVIEW ERS fluorescence microscope with GFP-actin staining, with a spatial resolution of $300 \times 350 \times 230$ voxels and a voxel size of $0.126 \times 0.126 \times 0.126\,\mu$m. The dataset contains 30 time points, with the first two sequences recorded at 2-minute intervals and the latter two at 20-second intervals. The second dataset is a collected *C.elegans* dataset with mCherry-stained cell membranes (Zhao et al., 2024), consisting of 6 groups with a spatial resolution of $256 \times 356 \times 160$ voxels and a voxel size of $0.18 \times 0.18 \times 0.18\,\mu$m. Each group comprises 30 time points with a 10-second interval. To reduce the inherent noise present in fluorescence images, both datasets were denoised using the BM4D algorithm (with globally standard deviation set as sigma parameters). The denoised results are denoted as $A549^{\text{hr}}$ and $CE^{\text{hr}}$, respectively. For self-supervised training of *SpatimeINR*, the original data are subjected to a $2\times$ downsampling and random scaling from the uniform distribution $U(2, 5)$, thereby constructing the data pairs $\{A549^{\text{hr}}_{(1/U(2))}, A549^{\text{lr}}_{(1/U(2,5))}\}$ and $\{CE^{\text{hr}}_{(1/U(2))}, CE^{\text{lr}}_{(1/U(2,5))}\}$. The original datasets $A549^{\text{hr}}$ and $CE^{\text{hr}}$ serve as the test sets to evaluate the reconstruction performance under out-of-distribution conditions.

### 4.2 IMPLEMENTATION DETAILS

The proposed *SpatimeINR* framework is implemented on the PyTorch platform, with details inspired by the open-source codes of LIIF (Chen et al., 2021b), RCAN (Chen et al., 2021a), CuNeRF (Chen et al., 2023), and SVIN (Guo et al., 2020). Initially, for a given target point, a 4D hypercube of radius 0.5 is centered at that point, and $N = 16$ coordinate points are uniformly sampled to obtain a coarse estimation of the volume density. The encoder adopts a RCAN-based architecture. The low-resolution input with shape $(B, X, Y, Z, N)$ is processed through a 3D convolutional layer followed

by several residual groups (each comprising RCAB modules and channel attention mechanisms) to extract high-dimensional features. These features are then mapped to the target channel number $T$ via a $1 \times 1 \times 1$ convolution, thereby capturing local spatiotemporal information. The latent code vector has a dimensionality of $|z| = 128$. Simultaneously, the implicit representation module utilizes an MLP structure analogous to NeRF, consisting of 8 fully connected hidden layers with 256 neurons each and ReLU activation functions. Notably, the center coordinate information is fused at the 4th layer to facilitate accurate kernel weight generation (details in Appendix F). During training, the model is optimized using the Adam optimizer (with a weight decay of $10^{-6}$) and a batch size of 2048, for a maximum of 20,000 iterations. The learning rate is linearly annealed from an initial value of $2 \times 10^{-3}$ to $2 \times 10^{-5}$. On a single NVIDIA RTX4090 GPU, training on volumes of size $(15, 128, 178, 80)$ requires approximately 20,000 iterations (around 20 minutes, Appendix H). In the inference phase, adjusting the sampling density facilitates the production of outputs with varying resolutions along different dimensions, thus fulfilling the requirements for high-quality, arbitrary-scale 4D spatiotemporal super-resolution reconstruction.

## 4.3 BENCHMARK

| | Method | ×2 | | | | ×4 | | | | ×6 | | | |
|---|---|---|---|---|---|---|---|---|---|---|---|---|---|
| | | PSNR | LPIPS | DSC | BSA | PSNR | LPIPS | DSC | BSA | PSNR | LPIPS | DSC | BSA |
| **A549** | Cubic* | 31.05 | 1.67 | 0.937 | 8.967 | 29.10 | 1.73 | 0.907 | 27.987 | 26.73 | 2.57 | 0.841 | 66.811 |
| | B-splines | 30.12 | 1.63 | 0.947 | 6.047 | 28.02 | 1.65 | 0.886 | 36.946 | 27.50 | 2.35 | 0.822 | 78.956 |
| | DynINR | 31.46 | 1.87 | 0.925 | 5.869 | 30.79 | 1.87 | 0.918 | 13.629 | 29.33 | 1.98 | 0.901 | 66.811 |
| | VoxelMorph | 31.18 | 1.42 | 0.947 | 5.170 | 31.91 | 1.90 | 0.917 | 18.023 | 30.16 | 2.28 | 0.919 | 57.511 |
| | UVI-Net | 33.88 | 1.41 | 0.943 | 5.160 | 32.17 | 1.64 | 0.928 | 7.487 | 30.52 | 1.76 | 0.920 | 22.308 |
| | SVIN | 33.39 | 1.60 | 0.928 | 4.909 | 31.83 | 1.61 | 0.938 | 7.607 | 30.25 | 1.88 | 0.916 | 14.599 |
| | **SpatimeINR** | **34.12** | **1.15** | **0.954** | **3.535** | **33.78** | **1.19** | **0.947** | **4.708** | **31.68** | **1.21** | **0.944** | **10.553** |
| **CE** | Cubic* | 29.30 | 1.18 | 0.921 | 9.046 | 29.05 | 1.92 | 0.909 | 29.323 | 25.87 | 2.02 | 0.880 | 80.265 |
| | B-splines | 30.78 | 2.53 | 0.950 | 8.781 | 29.50 | 1.91 | 0.904 | 37.853 | 25.65 | 2.02 | 0.807 | 77.999 |
| | DynINR | 32.90 | 1.21 | 0.942 | 8.096 | 30.10 | 1.81 | 0.927 | 17.685 | 29.12 | 2.06 | 0.893 | 79.566 |
| | VoxelMorph | 31.90 | 1.83 | 0.959 | 7.934 | 30.10 | 1.52 | 0.929 | 14.733 | 30.68 | 2.21 | 0.917 | 46.120 |
| | UVI-Net | 33.48 | 1.50 | 0.959 | 4.926 | 31.06 | 1.75 | 0.937 | 11.065 | 30.44 | 1.73 | 0.931 | 15.162 |
| | SVIN | 33.81 | 1.44 | 0.969 | 5.721 | 30.91 | 1.64 | 0.958 | 12.163 | 31.51 | 2.14 | 0.946 | 19.520 |
| | **SpatimeINR** | **34.95** | **1.14** | **0.970** | **4.516** | **33.94** | **1.21** | **0.973** | **7.868** | **32.40** | **1.25** | **0.954** | **13.138** |

**Table 1:** Quantitative Comparison Results for Temporal Super-Resolution. Experiments were conducted at scale factors of ×2, ×4, and ×6. **Bold** values indicate the highest scores achieved for each evaluation metric. Note that SVIN is a supervised method, while the remaining methods are unsupervised. The unit of LPIPS and BSA are $10^{-1}$ and $10^{-3}$. Due to space limitations, SSIM data is stored in Appendix G.

| | Method | ×2 | | | | ×4 | | | | ×6 | | | |
|---|---|---|---|---|---|---|---|---|---|---|---|---|---|
| | | PSNR | SSIM | LPIPS | DSC | PSNR | SSIM | LPIPS | DSC | PSNR | SSIM | LPIPS | DSC |
| **A549** | Cubic | 31.55 | 0.956 | 1.68 | 0.925 | 29.43 | 0.955 | 1.97 | 0.921 | 27.14 | 0.938 | 1.99 | 0.903 |
| | CuNeRF | 30.11 | 0.962 | 1.73 | 0.955 | 30.02 | 0.954 | 1.95 | 0.941 | 29.91 | 0.926 | 2.28 | 0.922 |
| | ArSSR | 32.17 | 0.978 | 1.46 | **0.976** | 32.21 | **0.972** | 1.93 | **0.961** | 31.22 | 0.958 | 2.14 | 0.945 |
| | SAINT | **33.22** | **0.981** | 1.41 | 0.969 | 31.18 | 0.971 | 1.27 | 0.957 | 31.51 | **0.967** | 2.66 | **0.951** |
| | SpatimeINR | 32.92 | 0.976 | **1.09** | 0.973 | **32.79** | 0.965 | **1.21** | 0.945 | **32.06** | 0.961 | **1.29** | 0.917 |
| **CE** | Cubic | 30.81 | 0.979 | 1.38 | 0.937 | 29.83 | 0.937 | 1.69 | 0.887 | 27.33 | 0.931 | 2.19 | 0.855 |
| | CuNeRF | 31.48 | 0.965 | 1.48 | 0.958 | 31.77 | 0.948 | 1.61 | 0.911 | 28.16 | 0.937 | 1.97 | 0.899 |
| | ArSSR | 33.84 | 0.977 | 1.34 | 0.965 | **32.53** | **0.966** | 1.94 | **0.925** | **30.99** | 0.962 | 2.39 | 0.909 |
| | SAINT | 32.24 | 0.978 | 1.29 | **0.971** | 32.17 | 0.957 | 1.64 | 0.917 | 30.62 | **0.964** | 2.08 | 0.906 |
| | SpatimeINR | **33.95** | **0.989** | **1.13** | 0.935 | 32.35 | 0.965 | **1.17** | 0.921 | 30.89 | 0.959 | **1.24** | **0.924** |

**Table 2:** Quantitative Comparison Results for Spatial Super-Resolution. Experiments were carried out at scale factors of ×2, ×4, and ×6. **Bold** values denote the highest scores achieved for each evaluation metric. Among these, ArSSR and SAINT are supervised methods, whereas the others employ unsupervised strategies. The unit of LPIPS is $10^{-1}$.

To comprehensively evaluate the performance of *SpatimeINR* for 3D and 4D data reconstruction, we designed two sets of comparative experiments. For temporal super-resolution, the competing methods include: improved Cubic (Ha et al., 2008), conventional B-splines interpolation (Eilers & Marx, 1996), DynINR (Feng et al., 2025), VoxelMorph (Balakrishnan et al., 2019), SVIN and UVI-Net. To reduce computational overhead, both the $A549^{\text{hr}}$ and $CE^{\text{hr}}$ datasets were downsampled in 3D by a factor of $1/2$, while the temporal dimension was preserved. Given that SVIN and UVI-Net's existing open-source implementations pertain only to the ACDC and 4D-Cardiac datasets, the corresponding methods were reimplemented in our study. Concurrently, the $A549^{\text{hr}}$ and $CE^{\text{hr}}$ datasets were rounded and partitioned into 28 groups, each represented as $\{I_{\text{front}}, I_{\text{interp}}, I_{\text{interp}}, I_{\text{interp}}, I_{\text{end}}\}$,

with the first 14 groups used for training and the latter 14 groups for evaluation. Other unsupervised methods were evaluated solely on the final two 4D groups. For spatial super-resolution experiments, comparisons were performed with the cubic method, ArSSR (Wu et al., 2022), SAINT (Peng et al., 2020), and CuNeRF (CycleINR (Fang et al., 2024) was excluded due to the unavailability of its source code). Among those compared, both the cubic method and CuNeRF are zero-shot approaches, whereas ArSSR and SAINT are supervised. In these experiments, the first two groups from the $A549^{\mathrm{hr}}$ and $CE^{\mathrm{hr}}$ datasets (each containing 60 low-resolution–high-resolution data pairs) were used for training, while the final two groups were designated for testing. The evaluation metrics include Peak Signal-to-Noise Ratio (PSNR) and Structural Similarity Index (SSIM) for quantitative assessment of interpolation accuracy. However, given the propensity of PSNR and SSIM to favor blurred images, the Learned Perceptual Image Patch Similarity (LPIPS) is additionally employed. Moreover, to further validate the effectiveness of the super-resolved outputs for downstream tasks, the MedSAM (Ma et al., 2024) is deployed for segmentation, and the Dice similarity coefficient (DSC) is computed based on the provided segmentation ground truth.

Distortions or blurring in critical regions of super-resolved images can severely compromise subsequent cell segmentation and quantitative analysis, and conventional metrics such as PSNR and DSC do not intuitively capture this issue. To quantify the effect of super-resolution reconstruction on preserving key morphological features in downstream biological tasks, we define a biological shape accuracy (BSA) metric based on sphericity (Figure 4 and (Wiesner et al., 2024; Zhao et al., 2024)). Specifically, for each 3D segmented shape at a given time point, the sphericity $\Phi$ is defined as

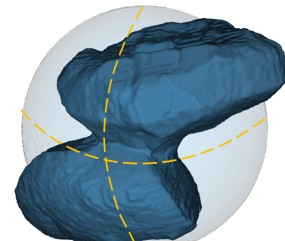

$$\Phi = \frac{\left(36\pi V^2\right)^{1/3}}{S},\tag{15}$$

**Figure 4:** Sphericity is defined in the range $(0, 1]$. An ideal sphere has a value of 1, and as a shape deviates from a sphere, its sphericity gradually declines.

where $V$ denotes the volume and $S$ the surface area of the segmented object. We then form sphericity vectors of length $t$ for the ground truth morphology (GT) and the super-resolved result (SR), denoted by $\hat{\Phi}^{(\mathrm{GT})} = \left[\Phi_1^{(\mathrm{GT})}, \Phi_2^{(\mathrm{GT})}, \dots, \Phi_t^{(\mathrm{GT})}\right]$ and $\hat{\Phi}^{(\mathrm{SR})} = \left[\Phi_1^{(\mathrm{SR})}, \Phi_2^{(\mathrm{SR})}, \dots, \Phi_t^{(\mathrm{SR})}\right]$, respectively. The mean squared error (MSE) between these vectors defines the BSA metric:

$$\mathrm{BSA} = \frac{1}{t}\sum_{i=1}^{t}\left(\Phi_i^{(\mathrm{GT})} - \Phi_i^{(\mathrm{SR})}\right)^2.\tag{16}$$

This metric effectively reflects the fidelity of super-resolution reconstruction in preserving critical structural features, thereby providing a robust basis for subsequent cell segmentation and quantitative analysis. A smaller BSA indicates a closer resemblance between the super-resolved and true morphological structures, while a larger value suggests significant structural deviations.

## 5 RESULTS

### 5.1 TEMPORAL SUPER-RESOLUTION

Table 1 presents the quantitative comparison results for temporal super-resolution of 4D data. Experimental results on both the $A549^{\mathrm{hr}}$ and $CE^{\mathrm{hr}}$ datasets indicate that *SpatimeINR* outperforms current state-of-the-art methods across all evaluation metrics and it significantly exceeds the performance of conventional non-learning approaches. Figure 5a illustrates the SR results on the $CE$ dataset at a 4× scale. The cubic method introduces filamentous artifacts, which lead to conspicuous segmentation errors in cell shape reconstruction tasks based on cell membrane signals. In contrast, the B-splines method exhibits pronounced deformations when processing non-linear and non-periodic motion data, thereby impeding the fluorescence signals in temporal imaging from accurately reflecting the true developmental changes during embryogenesis and resulting in erroneous reconstruction morphologies in subsequent segmentation tasks. Owing to the lack of encoder (Xie et al., 2025), DynINR effectively achieves spatiotemporal SR, its capability to capture the intricate dynamics of 4D spatiotemporal data is constrained. Its temporal interpolation manifests as predominantly linear, thereby inadequately reflecting the nonlinear biological phenomena

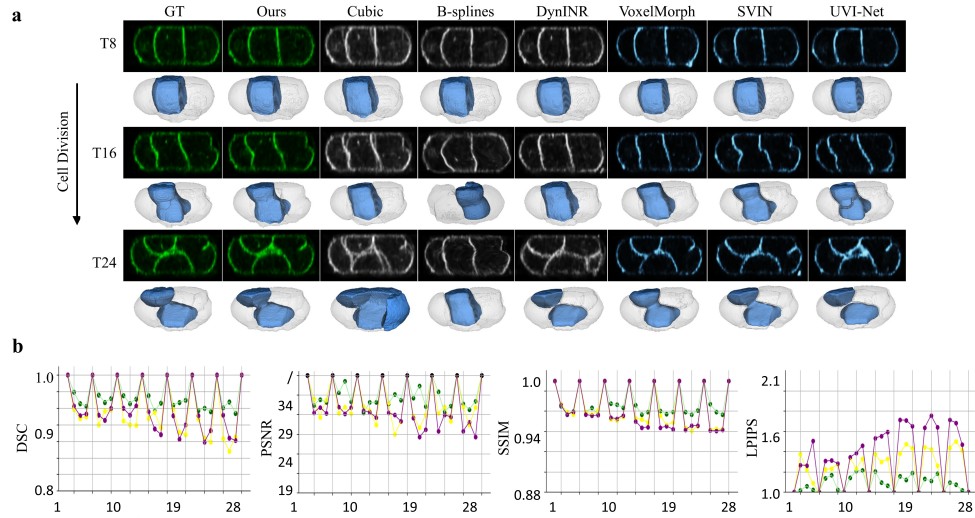

**Figure 5:** (a) Interpolation and downstream segmentation results on the CE dataset under $4\times$ scale, highlighting rapid cell division between Tp16 and Tp24; (b) Temporal evolution of evaluation metrics, with SpatimeINR, SVIN and UVI-Net representing the three best-performing groups.

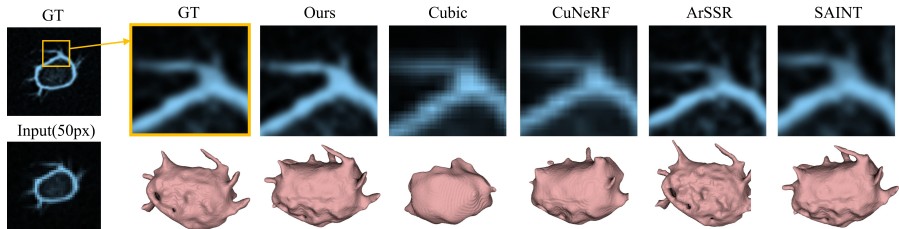

**Figure 6:** Visual Comparison of Spatial Super-Resolution Results. Experiments were conducted on the A549 dataset with 6× downsampling. The right panel presents the refined local region along with the corresponding downstream segmentation results, demonstrating the method's effectiveness in preserving fine details.

observed in practice. VoxelMorph, SVIN, and UVI-Net require independent processing for each set $\{I_{\text{front}}, I_{\text{interp}}, I_{\text{interp}}, I_{\text{interp}}, I_{\text{end}}\}$, thereby failing to fully capture the spatiotemporal dynamics in long 4D sequences. This leads to inferior reconstruction scores and deformation in rapidly changing regions, compounded by a complex processing pipeline that does not offer an integrated solution (Appendix I). Figure 5b displays the distributions of PSNR, SSIM, and DSC at a $4\times$ scale, demonstrating that the interpolation results of *SpatimeINR* consistently outperform those of SVIN and UVI-Net throughout the entire sequence. Furthermore, in regions where rapid changes occur due to cell division, the degradation in *SpatimeINR*'s performance is substantially less pronounced than that of SVIN and UVI-Net, validating that the spatiotemporal latent encoding method is more robust in capturing diverse biological motions and the 4D data manifold.

## 5.2 SPATIAL SUPER-RESOLUTION

Figure 6 presents qualitative comparisons for 3D volume spatial super-resolution. Using down-sampled lung cancer cell images at a 50px resolution as input, *SpatimeINR* reconstructs images that most closely approximate the ground truth, as further validated by downstream segmentation tasks. In contrast, the cubic method exhibits pronounced jagged edges during large-scale high-resolution recovery, leading to a loss of cellular morphological details, while the CuNeRF approach produces super-resolved outputs with blurred boundaries that hinder precise delineation of vector structures, result-

| INR | Rendering | CCL | latnet code | PSNR | BSA | LPIPS |
|---|---|---|---|---|---|---|
| ✓ | | | | 25.834 | 37.79 | 2.951 |
| ✓ | ✓ | | | 30.114 | 22.68 | 1.655 |
| ✓ | ✓ | ✓ | | 30.862 | 14.54 | 1.582 |
| ✓ | ✓ | ✓ | ✓ | **33.358** | **4.41** | **1.212** |

**Table 3:** Quantitative results for different module retention configurations. The leftmost four columns indicate the retention status of the corresponding modules: *INR*, *latnet code*, *Rendering*, and *CCL*. The unit of LPIPS and BSA are $10^{-1}$ and $10^{-3}$.

ing in incomplete recovery of cellular details. In conjunction with the quantitative evaluations presented in Table 2, these findings demonstrate that *SpatimeINR* significantly outperforms traditional methods and other unsupervised techniques in spatial super-resolution tasks, achieving performance comparable to that of supervised approaches reliant on extensive training data, thereby exhibiting state-of-the-art spatial super-resolution capabilities.

## 5.3 ABLATION STUDY

To comprehensively assess the contribution of each module to overall model performance, we conducted ablation studies on the spatiotemporal latent encoding, the 4D rendering, and the cycle consistency loss. The complete model achieved the best performance across evaluation metrics (Table 3). The baseline INR resulted in suboptimal reconstruction, exhibiting loss of image details, chessboard artifacts among discrete pixels (Figure 7), and the lowest scores across all quantitative measures. Incorporating the 4D rendering significantly improved continuity and precision; however, noticeable grid artifacts and minor noise persisted at the hypercube sampling boundaries. The addition of the cycle consistency module effectively reduced noise levels and maintained brightness uniformity, while providing an indirect constraint for large-scale upsampling in the absence of corresponding ground truth, thus enhancing overall reconstruction fidelity. Nonetheless, the model's capacity to capture nonlinear motion remained limited, supporting only linear interpolation of intermediate dynamics. With the integration of spatiotemporal latent encoding, the model's ability to perceive the 4D data manifold was substantially enhanced, enabling effective learning of true biological motion trajectories along the temporal axis. For instance, the *SpatimeINR* reconstruction in Figure 7 accurately reflected the rotational dynamics of the major axis (as indicated by the blue arrow) during human lung cancer cell metastasis. Quantitative evaluations across different latent dimensionalities (Appendix D) reveal that the reconstruction quality remains stable, whereas the absence of fundamental motion structures in low-resolution temporal data prevents *SpatimeINR* from recovering accurate super-resolution results (Appendix E).

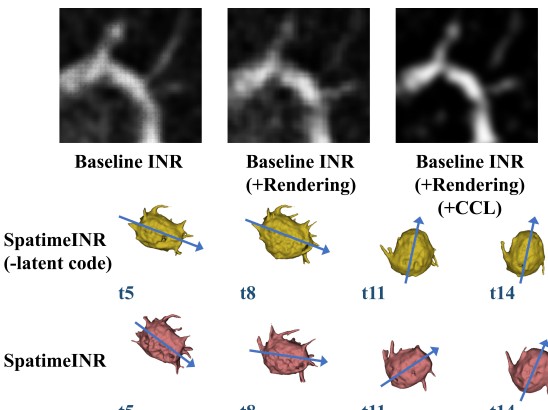

**Figure 7:** Visualization of the ablation study results. The fluorescence images depict the visual quality of the reconstructed images, while the cell shape illustrate the model's reconstruction of spatiotemporal features.

## 6 CONCLUSION

This paper introduces *SpatimeINR*, a 4D arbitrary-scale super-resolution reconstruction method based on implicit neural representation. By constructing continuous implicit representations from low spatiotemporal resolution 4D microscopic images and employing spatiotemporal latent variable encoding with volumetric rendering, the method accurately recovers the motion trajectories of biological samples while significantly reducing the operational complexity compared to traditional 4D interpolation techniques. This enhanced precision facilitates quantitative analysis and dynamic monitoring of the microscopic world, thereby advancing research on cellular dynamics. Extensive experiments on A549 lung cancer cells and *C. elegans* fluorescence datasets demonstrate that *SpatimeINR* outperforms existing methods in PSNR, SSIM, LPIPS, Dice, and BSA metrics. However, its computational cost and reconstruction quality are challenged in complex scenarios with insufficient motion structures, high noise, or ultra-large-scale 4D data. Future work will focus on more efficient modeling and optimization strategies.

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

# APPENDIX

## A    REPRODUCIBILITY STATEMENT

A complete proof of the proposed method is available in the Appendix J. The specific implementation details of *SpatimeINR* are shown in 4.2, and the benchmarks setting are shown in 4.3

## B    LARGE LANGUAGE MODELS USAGE STATEMENT

Large Language Models were only used to aid or polish writing.

## C    COMPLEX DYNAMIC CHANGES IN RECONSTRUCTION RESULTS

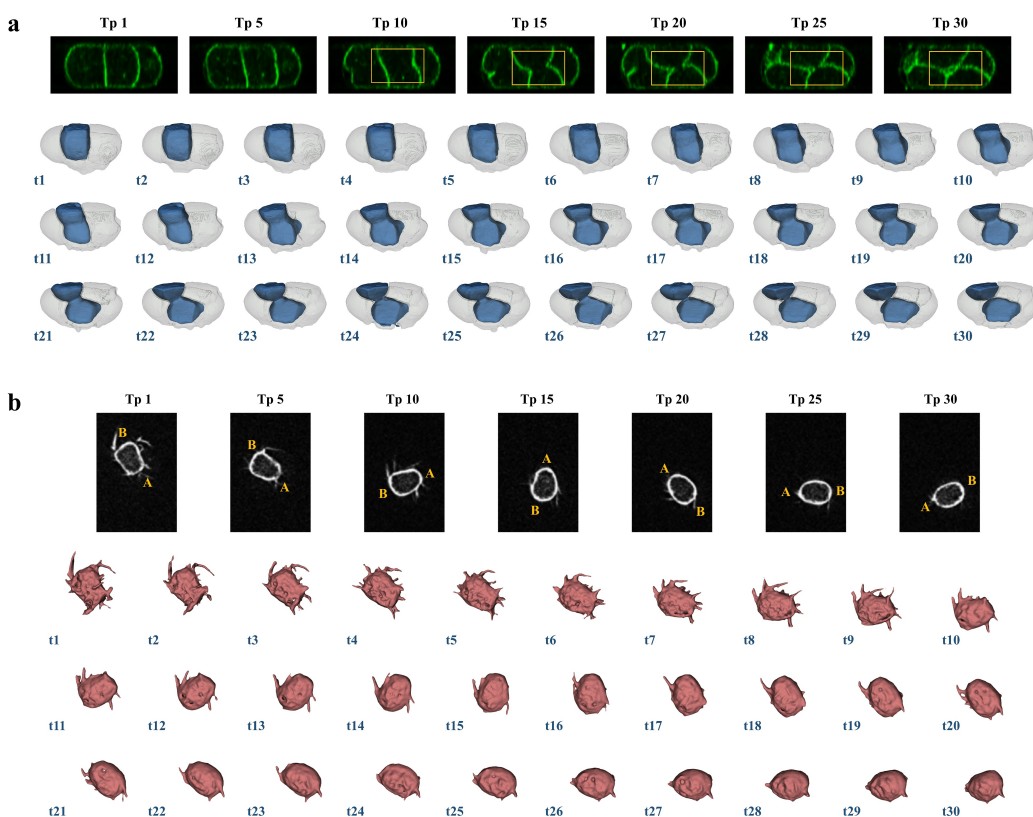

**Figure S1:** 3D visualization of complete reconstruction and downstream segmentation results obtained by SpatimeINR on (a) *C. elegans* embryonic development data and (b) human lung cancer cell metastasis data. Fluorescence images are shown at intervals of 4 to clearly reveal dynamic changes while reducing visual redundancy.

*SpatimeINR* faithfully reproduces the division process of the early ABp cell in *C. elegans* embryogenesis (the four-cell stage, wherein ABp eventually gives rise to the nervous system and portions of the epidermis) into ABpl and ABpr, as highlighted by the yellow box in Figure S1a. From time points $t_1$–$t_{10}$, chromatin condenses into chromosomes and cell division occurs without marked deformation. Between $t_{11}$ and $t_{20}$, cytokinesis accelerates with a rapidly contracting cleavage furrow, and from $t_{21}$ to $t_{30}$ the furrow further narrows at a decelerating pace until two distinct cells emerge.

The reconstruction of human lung cancer cell metastasis data reveals amoeboid migration during tumor progression and dissemination. This phenomenon is characterized by the disappearance of

pseudopodia, the attainment of a smooth cell surface, and rapid movement. In Figure S1a, vector AB in the fluorescence image denotes the cell's major axis in the two-dimensional cross-section; its swift rotation signals vigorous motility during metastasis. From $t_1$ to $t_{15}$, lung cancer cells exhibit retraction of cell margins, loss of pseudopodia (the spiky surface structures), and reduced adhesion, transforming into a rounded morphology that facilitates passage through narrow tissue gaps. From $t_{16}$ to $t_{30}$, rapid cellular movement coupled with tissue infiltration may portend further tumor deterioration and metastasis.

Both processes display pronounced nonlinear and irreversible dynamics. Moreover, precise observations of embryonic development and cancer cell dispersal symmetrically expose the dynamics of life's inception and apoptosis; the brief interstitial phase between them encapsulates the unassuming yet remarkable manifestation of life.

## D THE EFFECT OF LATENT DIMENSIONALITY ON SPATIOTEMPORAL FIDELITY,

| Latent Dim | 0 | 32 | 64 | 128 | 256 |
|---|---|---|---|---|---|
| PSNR | 30.862 | 31.654 | 32.286 | 33.358 | 33.251 |
| BSA ($\times 10^{-3}$) | 14.54 | 6.50 | 4.93 | 4.708 | 5.03 |
| Time (min) | 16 | 20 | 20 | 22 | 27 |

**Table S1:** Reconstruction outcomes, morphological accuracy, and training time of SpatimeINR under different latent dimensionalities.

We investigated the impact of the latent dimension on spatiotemporal fidelity by varying only the latent code dimension while keeping the remainder of the network unchanged. The latent dimension was set to 0, 32, 64, 128, and 256, and reconstruction performance on the CE dataset at $4\times$ scale was evaluated using PSNR, Biological Shape Accuracy (BSA, Section 4.3), and training time. Results show that without latent code (dimension 0), the model required 16 minutes to train but yielded substantially worst PSNR and BSA. At a dimension of 32, both PSNR and BSA are better, with training time increasing to 20 minutes. When the dimension ranged from 64 to 256, the PSNR and BSA metrics remained stable, with training time reaching 27 minutes at 256 dimensions. Although the training time increased progressively with the latent dimension, its benefit to reconstruction quality was marginal.

## E FAILURE CASES

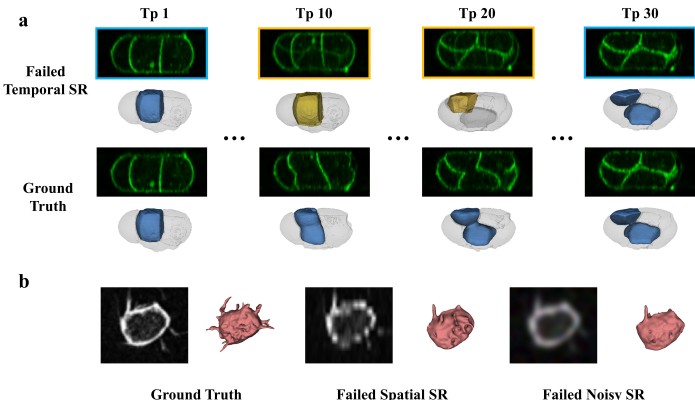

**Figure S2:** Failure cases for SpatimeINR. (a) illustrates a failure instance in the temporal super-resolution task, where the blue boxes indicate the input start and end frames and the yellow box displays the predicted intermediate frame exhibiting significant artifacts that cause downstream segmentation tasks to fail. (b) presents the reconstruction result under highly anisotropic conditions with severe Gaussian noise ($\sigma = 100$), leading to pronounced detail loss.

| Temporal Context Length | 30 | 15 | 10 | 5 | 2 | 1 |
|---|---|---|---|---|---|---|
| PSNR | 36.613 | 34.12 | 32.562 | 31.204 | 23.41 | 19.88 |
| BSA ($\times 10^{-3}$) | 2.623 | 3.125 | 3.99 | 4.09 | 64.387 | 97.756 |

**Table S2:** Variations in SpatimeINR reconstruction outcomes under different input temporal context lengths.

| Noise Level ($\sigma$) | 0 | 5 | 10 | 30 | 50 | 100 |
|---|---|---|---|---|---|---|
| PSNR | 33.764 | 33.358 | 32.84 | 29.939 | 27.83 | 25.522 |
| BSA ($\times 10^{-3}$) | 3.571 | 3.581 | 6.796 | 33.821 | 47.266 | 78.619 |

**Table S3:** Impact of varying levels of added Gaussian noise on SpatimeINR reconstruction outcomes.

We supplemented our study with a systematic investigation of the effect of temporal context length during encoding. *SpatimeINR* was trained with input sequences of length $t = 30$ (SR factor 1), $t = 15$ (2×), $t = 10$ (3×), $t = 5$ (6×), $t = 2$ (15×), and $t = 1$ (30×), and then used to predict the complete temporal sequence at $t = 30$. Experimental results indicate that when $t \geq 5$, the overall performance remains stable despite slight declines in prediction accuracy. In contrast, at $t = 2$ or $t = 1$, the input data loses its basic motion structure on the temporal axis, leading to a marked failure in super-resolution reconstruction. This behavior is analogous to a microscope being unable to capture a complete motion process under very rapid motion. To further simulate scenarios with severe anisotropy, we conducted a 10-fold downsampling experiment along the $z$-axis. The results reveal a significant deterioration in reconstruction quality (Figure S2ab).

Concurrently, we evaluated model robustness by adding Gaussian noise with $\sigma = 0$ (noise-free), 5 (mild), 10 (mild), 30 (moderate), 50 (severe), and 100 (severe) to the images. The experiments show that *SpatimeINR* is robust to mild noise, with negligible changes in performance; however, under moderate or higher noise levels, both PSNR and BSA deteriorate considerably, and the preservation of fine details in downstream segmentation tasks is greatly diminished (Figure S2b). These findings suggest that moderate denoising is advisable in practical applications.

# F   THE ARCHITECTURE OF ENCODER AND MLP

We employ the RCAN4D architecture with the following implementation. The input is a 5D tensor of shape $(bsize, X, Y, Z, T)$. After permuting its dimensions to $(bsize, T, X, Y, Z)$, the tensor is fed to a head module composed of a 3D convolution (default_conv3d) with symmetric padding for preliminary feature extraction. Deep features are then extracted by a main module consisting of multiple ResidualGroup3D blocks. Each ResidualGroup3D contains several RCAB3D modules; within each RCAB3D, two layers of 3D convolution with ReLU activation are applied alongside a CALayer3D—which uses adaptive average pooling and a $1 \times 1 \times 1$ convolution to implement channel attention—with the input features combined via a residual connection. In the tail module, a $1 \times 1 \times 1$ convolution maps the features to a low-dimensional latent code of dimension len_Z, and the result is passed through a tanh function (scaled by $\pi$) for output.

For the conditional MLP, our revised manuscript details its implementation as follows. The MLP encodes input coordinates at multiple scales using a preset maximum frequency along with sine and cosine functions (Equation 3), yielding an embedding vector. This encoding is concatenated with additional channel information (the sampling results from Section 3.2) to form the complete input, which is then processed by a network of fully connected layers. The MLP comprises eight hidden layers, each with 256 neurons and ReLU activation. In the fourth layer, a predefined skip connection concatenates the complete input with the current output to reinforce the recovery of high-frequency details. Finally, a linear layer produces the target output dimension.

# G  SSIM VALUES IN THE QUANTITATIVE COMPARISON RESULTS FOR TEMPORAL SUPER-RESOLUTIO

|  | Method | ×2 SSIM | ×4 SSIM | ×6 SSIM |
|---|---|---|---|---|
| **A549** | Cubic* | 0.933 | 0.918 | 0.820 |
|  | B-splines | 0.937 | 0.886 | 0.856 |
|  | DynINR | 0.931 | 0.954 | 0.925 |
|  | VoxelMorph | 0.953 | 0.953 | 0.959 |
|  | UVI-Net | 0.961 | 0.974 | 0.950 |
|  | SVIN | 0.969 | 0.967 | 0.943 |
|  | **SpatimeINR** | **0.972** | **0.978** | **0.962** |
| **CE** | Cubic* | 0.910 | 0.907 | 0.840 |
|  | B-splines | 0.940 | 0.898 | 0.832 |
|  | DynINR | 0.951 | 0.939 | 0.901 |
|  | VoxelMorph | 0.949 | 0.958 | 0.953 |
|  | UVI-Net | 0.979 | 0.963 | 0.960 |
|  | SVIN | 0.979 | 0.974 | 0.953 |
|  | **SpatimeINR** | **0.985** | **0.976** | **0.971** |

**Table S4:** SSIM values in the Quantitative Comparison Results for Temporal Super-Resolutio.

# H  COMPUTE REPORT

| Dataset | Train | Inference (×2) | Inference (×4) | GPU Memory |
|---|---|---|---|---|
| CE (t=15) | 22 | 18 | 35 | 4000 |
| A549 (t=15) | 21 | 16 | 33 | 4000 |
| ACDC (t=19) | 21 | 20 | 35 | 5000 |
| Syn (t=240) |  | CUDA out of memory |  |  |
| Syn* (t=240) | 55 | 22 | 39 | 9000 |

**Table S5:** Computation report of SpatimeINR on diverse datasets, including two synthetic ultra-large datasets.

We systematically evaluated five four-dimensional datasets on an NVIDIA RTX 4090. The CE dataset has volume dimensions of $(15, 128, 178, 80)$ (ordered as $t, x, y, z$), the A549 dataset is $(15, 150, 175, 115)$, and the ACDC (cardiac MRI) dataset is $(15, 180, 224, 10)$. Additionally, by resizing eight groups (each comprising 30 time points) of the CE and A549 datasets to match the CE dimensions and concatenating them along the temporal axis, we constructed a large-scale dataset with volume dimensions of $(240, 128, 178, 80)$ for evaluating *SpatimeINR* on extensive data.

Experimental results indicate that training and inference at 2× scale require approximately 20 minutes per dataset, while under 4× super-resolution the inference time stays below 40 minutes, with an almost linear increase in processing time. After optimizing with `torch.cuda.empty_cache()`, the peak GPU memory usage remains under 5000 MB, averaging around 4000 MB, which demonstrates that the method runs efficiently on most GPUs. For the synthetic dataset Syn, the original data size caused CUDA memory shortages. By down-sampling the $x$, $y$, and $z$ axes by a factor of two while preserving temporal resolution, we obtained a dataset, denoted Syn*, with volume dimensions of $(240, 64, 89, 40)$; its training time is approximately 55 minutes, and both inference time and memory consumption are within acceptable limits. These results suggest a practical strategy for processing larger four-dimensional data by reducing volume resolution—with care to preserve critical information—to lower training time and memory load.

# I    BENCHMARK RESULTS OF SPATIMEINR COMPARED TO UVI-NET ON THE CARDIAC (ACDC) AND CE DATASETS.

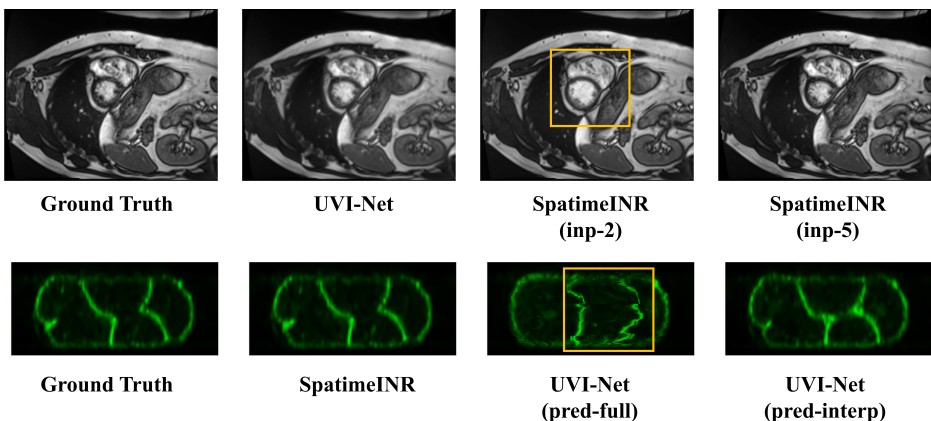

| | | | |
|---|---|---|---|
| **Ground Truth** | **UVI-Net** | **SpatimeINR**
**(inp-2)** | **SpatimeINR**
**(inp-5)** |
| **Ground Truth** | **SpatimeINR** | **UVI-Net**
**(pred-full)** | **UVI-Net**
**(pred-interp)** |

**Figure S3:** Qualitative benchmark comparison of *SpatimeINR* and UVI-Net on the ACDC and CE datasets. The first row (ACDC dataset) shows *SpatimeINR*'s prediction of an erroneous intermediate frame (yellow box) using only the start and end frames. The second row (CE dataset) displays UVI-Net's prediction of an erroneous intermediate frame (yellow box) using only the start and end frames.

| Dataset | Metric | UVI-Net | SaptimeINR (inp-2) | SpatimeINR (inp-5) |
|---|---|---|---|---|
| ACDC | PSNR | 33.59 | 28.85 | 34.01 |
| | SSIM | 0.978 | 0.91 | 0.978 |
| | LPIPS | 1.066 | 4.428 | 1.012 |

**Table S6:** Quantitative results of UVI-Net and *SpatimeINR* under different input conditions on the ACDC dataset. The unit of LPIPS is $10^{-2}$

We designed several comparative experiments. On the ACDC dataset, one scheme provided *SpatimeINR* only with the first and last frames, while another scheme input five frames (the initial frame, frames at 25%, 50%, and 75% of the sequence, and the final frame) to predict intermediate motion over 10 test samples. UVI-Net was pretrained on 90 datasets and tested on 10 samples in accordance with the strategy described by its authors. Results show that when predicting intermediate motion solely from the boundary frames, *SpatimeINR* failed to recover the omitted motion information (Table S6, Figure S3), with its PSNR, SSIM, and LPIPS significantly below those of UVI-Net. However, when additional intermediate frames provided a basic motion structure, *SpatimeINR* markedly improved its reconstruction of the non-periodic cardiac motion and outperformed UVI-Net across all metrics. Given the disparity in input information, these comparisons remain only indicative.

| Dataset | Metric | SaptimeINR | UVI-Net (pred-full) | UVI-Net (pred-interp) |
|---|---|---|---|---|
| CE | PSNR | 32.4 | 23.33 | 30.44 |
| | SSIM | 0.971 | 0.762 | 0.96 |
| | LPIPS | 1.25 | 3.946 | 1.73 |

**Table S7:** Quantitative results of *SpatimeINR* and UVI-Net under different input conditions on the CE dataset. The unit of LPIPS is $10^{-1}$

Furthermore, supplementary comparisons were conducted on the CE dataset, which features complex and irreversible nonlinear variations. In one configuration, UVI-Net employed a self-supervised training protocol consistent with the original paper to predict 28 intermediate frames from the boundary frames only. In another configuration, UVI-Net was pretrained on 12 pairs of boundary data formed through grouping ($\{I_{\text{front}}, I_{\text{interp}}, I_{\text{interp}}, I_{\text{interp}}, I_{\text{end}}\}$) and then predicted 3 intermediate frames per group. Meanwhile, *SpatimeINR* directly used 5 low-temporal-resolution frames to predict the complete 30-frame high-temporal-resolution sequence. Experimental results indicate that

UVI-Net exhibited noticeable prediction errors with only boundary inputs (Table S7, Figure S3); although grouped prediction yielded some improvement, its complexity and heavy preprocessing render it impractical for real microscopic imaging scenarios.

In summary, UVI-Net is suitable for predicting motion trajectories from boundary frames in tasks characterized by modest, simple variations (*e.g.*, medical image registration or MRI), whereas *SpatimeINR* addresses 4D microscopic data with complex dynamics in long sequences, offering a more effective and practical solution.

## J  MATHEMATICAL DERIVATION FROM 1D VOLUME RENDERING TO 4D SPATIOTEMPORAL VOLUME RENDERING

Consider a ray parameterized as

$$\mathbf{r}(t) = \mathbf{o} + t\,\mathbf{d}, \quad t \in [t_n, t_f], \tag{17}$$

where $\mathbf{o}$ is the camera origin and $\mathbf{d}$ is the ray direction. Along this ray, the continuous volume rendering equation is defined by

$$C(\mathbf{r}) = \int_{t_n}^{t_f} T(t)\,\sigma\big(\mathbf{r}(t)\big)\,c\big(\mathbf{r}(t), \mathbf{d}\big)\,dt, \tag{18}$$

where: $\sigma\big(\mathbf{r}(t)\big)$ denotes the volume density at point $\mathbf{r}(t)$, which governs light absorption and scattering; $c\big(\mathbf{r}(t), \mathbf{d}\big)$ denotes the emitted color (or radiance) at point $\mathbf{r}(t)$ in direction $\mathbf{d}$; The transmittance $T(t)$ is given by the Beer–Lambert formulation:

$$T(t) = \exp\left(-\int_{t_n}^{t} \sigma\big(\mathbf{r}(s)\big)\,ds\right). \tag{19}$$

For practical numerical implementation, the interval $[t_n, t_f]$ is discretized into $N$ segments with sample positions $\{t_i\}_{i=1}^{N}$ and corresponding step sizes $\Delta t_i = t_{i+1} - t_i$. Defining

$$\alpha_i = 1 - \exp(-\sigma_i\,\Delta t_i), \quad T_i = \exp\left(-\sum_{j=1}^{i-1} \sigma_j\,\Delta t_j\right), \tag{20}$$

where $\sigma_i = \sigma\big(\mathbf{r}(t_i)\big)$ and $c_i = c\big(\mathbf{r}(t_i), \mathbf{d}\big)$, the discrete approximation is given by

$$C(\mathbf{r}) \approx \sum_{i=1}^{N} T_i\,\alpha_i\,c_i. \tag{21}$$

This equation accumulates the weighted contributions of each sampled point along the ray, where the weights are determined by the cumulative transmittance and local absorption.

Extending the rendering paradigm from 1D to 4D (*i.e.*, three spatial dimensions plus time) provides two critical advantages: 1: The rendering approach defines a continuous mapping from spatial–temporal coordinates to color and density. This allows for seamless recovery of both the detailed spatial structure and the dynamic temporal evolution, which is particularly important for applications such as live-cell imaging. 2: Due to physical constraints (*e.g.*, phototoxicity, exposure limitations), high-resolution 4D data are typically under-sampled. By using a continuous implicit representation as in volume rendering, interpolations across both space and time are guided by physically meaningful accumulation, reducing interpolation artifacts and blockiness. To extend the 1D rendering formulation to 4D spatiotemporal data, we follow these steps:

Let the target query point be

$$\hat{\boldsymbol{x}}_{st} = (\hat{x}, \hat{y}, \hat{z}, \hat{t}) \in [-1, 1]^4. \tag{22}$$

Around $\hat{\boldsymbol{x}}_{st}$, we define a 4D hypercube with edge length $l$ by

$$B_4\left(\hat{\boldsymbol{x}}_{st}, \frac{l}{2}\right) = \left\{\boldsymbol{\mu} \in \mathbb{R}^4 \;\middle|\; \|\boldsymbol{\mu} - \hat{\boldsymbol{x}}_{st}\|_1 \leq \frac{l}{2}\right\}. \tag{23}$$

Although the $L_1$ norm is used to define the hypercube, for the purpose of isotropic integration, we transition to the Euclidean ($L_2$) framework when formulating the integration.

Within the hypercube $B_4$, we uniformly sample $M$ points, where typically $M = n^4$ for some positive integer $n$. Denote these sampled points by

$$\{\boldsymbol{x}_{st}^{(m)}\}_{m=1}^M, \quad \boldsymbol{x}_{st}^{(m)} = (x^{(m)}, y^{(m)}, z^{(m)}, t^{(m)}). \tag{24}$$

Each sampled point is then mapped to a higher-dimensional feature space via a position encoding function:

$$\gamma(\boldsymbol{x}_{st}^{(m)}) = \Big(\sin(2^0 \pi \, \boldsymbol{x}_{st}^{(m)}), \cos(2^0 \pi \, \boldsymbol{x}_{st}^{(m)}), \ldots, \sin(2^{L-1} \pi \, \boldsymbol{x}_{st}^{(m)}), \cos(2^{L-1} \pi \, \boldsymbol{x}_{st}^{(m)})\Big). \tag{25}$$

In conjunction with latent feature codes $z(\boldsymbol{x}_s)$ extracted from the spatial coordinates $\boldsymbol{x}_s = (x, y, z)$ by an encoder network, the concatenated vector $\big[\gamma(\boldsymbol{x}_{st}^{(m)}), z(\boldsymbol{x}_s^{(m)})\big]$ is input to a conditioned multi-layer perceptron (MLP) $f_\theta$. The MLP predicts the color $c^{(m)}$ and density $\sigma^{(m)}$ at each sample:

$$\big(c^{(m)}, \sigma^{(m)}\big) = f_\theta\Big(\gamma(\boldsymbol{x}_{st}^{(m)}), \, z(\boldsymbol{x}_s^{(m)})\Big). \tag{26}$$

We assume that the density within $B_4$ is isotropic in the sense that it depends only on the Euclidean distance from the query point $\hat{\boldsymbol{x}}_{st}$. Denote this distance by

$$r = \|\boldsymbol{x} - \hat{\boldsymbol{x}}_{st}\|. \tag{27}$$

In four-dimensional Euclidean space, the infinitesimal volume element in spherical coordinates is given by

$$dV = \omega_3 \, r^3 \, dr, \quad \text{where } \omega_3 = 2\pi^2 \tag{28}$$

is the surface area of the unit 3-sphere. By analogy with the 1D case and following the Beer–Lambert law, the transmittance from the center to a distance $r$ is defined as

$$T(r) = \exp\left(-\omega_3 \int_0^r s^3 \, \sigma(s) \, ds\right). \tag{29}$$

Thus, the continuous 4D volume rendering integral is formulated as

$$C\Big(\hat{\boldsymbol{x}}_{st}, l\Big) = \omega_3 \int_0^{r_{\max}} r^3 \, \sigma(r) \, c(r) \, T(r) \, dr, \tag{30}$$

where $r_{\max}$ is the maximum effective distance within the hypercube.

For numerical computation, the interval $[0, r_{\max}]$ is discretized into $M$ sub-intervals. Let the discrete radii be $\{r_i\}_{i=1}^M$ with step sizes $\Delta r_i = r_{i+1} - r_i$. Within each interval, define the local absorption coefficient by

$$\alpha_i = 1 - \exp(-\sigma_i \, \Delta r_i), \tag{31}$$

where $\sigma_i = \sigma(r_i)$ and $c_i = c(r_i)$. The cumulative transmittance from $r = 0$ to $r = r_i$ is approximated as

$$T_i = \exp\Big(-\omega_3 \sum_{j=1}^i \sigma_j \, r_j^3 \, \Delta r_j\Big). \tag{32}$$

Hence, the discrete 4D rendering formulation is given by

$$C\Big(\hat{\boldsymbol{x}}_{st}, l\Big) \approx \omega_3 \sum_{i=1}^M T_{i-1} \, \alpha_i \, c_i. \tag{33}$$

Substituting $\omega_3 = 2\pi^2$, we have

$$C\Big(\hat{\boldsymbol{x}}_{st}, l\Big) \approx 2\pi^2 \sum_{i=1}^M T_{i-1} \, \alpha_i \, c_i. \tag{34}$$

An alternative formal expression is

$$C\Big(\hat{\boldsymbol{x}}_{st}, l\Big) = 2\pi^2 \sum_{i=1}^M \frac{r_i^3 \Big(1 - \exp\big(-\sigma_i \, (r_{i+1} - r_i)\big)\Big) c_i}{\exp\Big(2\pi^2 \sum_{j=1}^i r_j^3 \, \sigma_j \, (r_{j+1} - r_j)\Big)}. \tag{35}$$

