# OpenReview forum: "Implicit Reconstruct Spatiotemporal Super-Resolution Microscopy in Arbitrary Dimension"
_ICLR.cc/2026/Conference — ICLR 2026 Conference Withdrawn Submission_

### Official Review · Reviewer_4zRG · 2025-10-27

**Soundness:** 3
**Presentation:** 3
**Contribution:** 2
**Rating:** 6
**Confidence:** 3

**Summary:**

This paper introduces SpatimeINR, a novel framework designed to address the challenges of low spatiotemporal resolution, phototoxicity, and anisotropy in 4D fluorescence microscopy. The method uses an implicit neural representation (INR) to achieve arbitrary-scale super-resolution for 4D data (3D space + time). The motivation is clear, and the experimental results are solid. However, the method has not introduced domain-specific priors from biological processing.

**Strengths:**

(1) The motivation is clear and makes sense.

(2). The paper is well-written and easy to follow.

(3) The experimental results effectively validate the effectiveness of proposed method.

**Weaknesses:**

(1) The paper focuses on "biological processing." However, the 4D INR modeling of cellular behavior does not appear to incorporate specific domain knowledge from biology. in fact, the proposed method is more like a general-purpose 4D SR approach.

(2) Similar to LIIF, the trained network is capable of outputting SR results directly without fitting each sample individually. If the extracted input features from the encoder can be reused when querying different upsampling scales for the same input? Authors should provide more detailed analysis regarding inference time and the reusability of features for arbitrary-scale super-resolution.

(3) Missing discussion with recent 2D ASR methods using explicit Gaussians.

[1] Generalized and Efficient 2D Gaussian Splatting for Arbitrary-scale Super-Resolution

[2] Pixel to Gaussian: Ultra-Fast Continuous Super-Resolution with 2D Gaussian Modeling

**Questions:**

See weakness.

---

> ### Author Response · Authors · 2025-11-25
> **Comment 14**
>
> **Q14: The paper focuses on "biological processing." However, the 4D INR modeling of cellular behavior does not appear to incorporate specific domain knowledge from biology. in fact, the proposed method is more like a general-purpose 4D SR approach.**
>
> Thank you very much for the valuable comments!
>
> > We acknowledge that our current model does not explicitly incorporate specific priors from the biological domain. Our design intent is to address two representative problems in 4D microscope time-series imaging: First, due to the anisotropic resolution inherent in microscope volumetric imaging, there is insufficient resolution along the z-axis; simultaneously, to avoid issues such as photobleaching caused by excessively high sampling density, the sampling density along the t-axis is limited. Second, live biological samples often exhibit nonlinear and non-periodic dynamic changes over time (**Appendix C, Figure S1**). Based on these two unique challenges, we propose the SpatimeINR method, which provides a super-resolution image enhancement approach for 4D microscope imaging at arbitrary scales through the design of local spatiotemporal feature encoding and a unified implicit 4D volumetric rendering framework.
>
> We have integrated these discussion into the revised manuscript’s **Introduction (Page 1/Line 27)**.

---

> ### Author Response · Authors · 2025-11-25
> **Comment 15**
>
> **Q15: Similar to LIIF, the trained network is capable of outputting SR results directly without fitting each sample individually. If the extracted input features from the encoder can be reused when querying different upsampling scales for the same input? Authors should provide more detailed analysis regarding inference time and the reusability of features for arbitrary-scale super-resolution.**
>
> Thank you very much for the valuable comments!
>
> > In SpatimeINR, we adopt a design similar to LIIF, where we perform the encoder feature extraction on the input low-resolution image only once, and then reuse the extracted feature map for all subsequent arbitrary-scale super-resolution queries.
> >
> > During both training and inference, for non-integer points in high-resolution sampling we use a nearest neighbor search method based on the L1 distance. Specifically, for any input spatial coordinate x_s in [-1,1]^3, we search in the low-resolution feature map for the grid point that is closest to x_s in terms of L1 distance. The corresponding latent code z(x_s) is taken as the extracted vector. This query process yields a latent vector of length d within the 3D space, which integrates both the spatial features at the (x,y,z,:) position in the current 4D data (x,y,z,t) and the temporal features from time 0 to t. Subsequently, we concatenate the query coordinates with the complete d-dimensional vector and input the result into an MLP.
> >
> > In **Appendix H**, we supplement our paper with inference times across different datasets, which demonstrate that the inference time grows approximately linearly. Considering that 4D data is inherently large, the inference time is longer compared to 2D or 3D images, yet it remains within an acceptable range. After optimization using `torch.cuda.empty_cache()`, the GPU memory peak does not exceed 5000 MB, averaging around 4000 MB, which demonstrates that the method can efficiently run on the vast majority of GPUs. In Section 5.1, we quantitatively compared the reliability of SpatimeINR’s sampling at different scales; experimental results show that SpatimeINR outperforms the current ASISR method and 4D image interpolation methods across various metrics.
>
> We have integrated these discussion into the revised manuscript’s **Section 3.2 (Page 4/Line 201) and Appendix H (Page 17/Line 888)**.

---

> ### Author Response · Authors · 2025-11-25
> **Comment 16**
>
> **Q16: Missing discussion with recent 2D ASR methods using explicit Gaussians.**
>
> Thank you very much for the valuable comments!
>
> > We highly appreciate the innovative methods proposed by Chen et al. [1] and Peng et al. [2] for 2D arbitrary-scale super-resolution based on Gaussian splatting. Like LIIF, these methods are poised to become benchmark techniques in the field of arbitrary-scale super-resolution. Chen's method employs 2D Gaussian splatting along with a differentiable, scale-aware rasterization approach to achieve high-quality and high-speed super-resolution. Peng's method, based on a deep Gaussian prior, introduces an arbitrary-scale upsampling Gaussian method that does not directly utilize implicit neural representations (INR). These two approaches, grounded in explicit Gaussian modeling, are primarily suited for conventional 2D images; their advantage lies in leveraging region-level Gaussian distributions to capture strong local texture details and deliver fast rendering performance.
> >
> > However, considering that both methods are relatively new—with Peng's method still under review—their applicability to biological data and medical images may require further experimental validation and optimization. For example, 4D time-lapse microscope imaging data exhibit distinct characteristics compared to conventional 2D images: noticeably low resolution along the z-axis, sparse data along the t-axis, and common nonlinear, non-periodic dynamic changes over time. The SpatimeINR method is designed specifically to address these challenges by utilizing spatiotemporal implicit representations and continuous 4D volumetric rendering, which not only restore high-resolution spatial details but also capture complex dynamic changes along the time axis. Although the methods by Chen [1] and Peng [2] have achieved promising results on conventional 2D images, aspects such as computational load, detail restoration, and the ability to recover nonlinear dynamic changes along the t-axis in biological experimental data still require further validation.
> >
> > [1] Generalized and Efficient 2D Gaussian Splatting for Arbitrary-scale Super-Resolution
> > [2] Pixel to Gaussian: Ultra-Fast Continuous Super-Resolution with 2D Gaussian Modeling
>
>
> We have integrated these discussion into the revised manuscript’s **Introduction (Page 1/Line 27)**.

---

### Official Review · Reviewer_yv2D · 2025-10-30

**Soundness:** 2
**Presentation:** 3
**Contribution:** 2
**Rating:** 4
**Confidence:** 4

**Summary:**

This paper introduces SpatimeINR, a model designed for 4D super-resolution of microscopy images. Unlike traditional methods that upscale images by fixed factors, SpatimeINR constructs an implicit neural representation (INR) over the spatiotemporal domain
(x,y,z,t). This enables continuous sampling at arbitrary spatial and temporal resolutions. The model consists of two core components: a spatiotemporal feature extractor and an INR renderer. It is trained using a reconstruction loss, with an additional cycle-consistency loss to enhance perceptual quality. Experimental results show that SpatimeINR achieves state-of-the-art (SOTA) performance on two microscopy benchmarks.

**Strengths:**

- The paper is clearly written and easy to follow.

- The proposed method enables arbitrary-scale 4D super-resolution.

- The model demonstrates SOTA performance on evaluated benchmarks.

**Weaknesses:**

- The method’s novelty is somewhat limited. The key components—INR-based super-resolution and cycle-consistency loss—are already well-studied, and the contribution mainly lies in combining established techniques and applying them to the microscopy domain.

- The baseline comparison appears outdated; for example, SVIN may not be the strongest reference point. It would be more convincing to compare against recent natural image super-resolution methods retrained on microscopy data (e.g., [1]).

- The experiments are performed on benchmarks different from those used by prior works, making it difficult to fairly assess whether SpatimeINR truly outperforms existing methods.

[1] Ekanayake, Mevan, et al. "SeCo-INR: Semantically Conditioned Implicit  Neural Representations for Improved Medical Image Super-Resolution." *2025 IEEE/CVF Winter Conference on Applications of Computer Vision (WACV)*. IEEE, 2025.

**Questions:**

- In ablation, instead of testing a combination of three different components, can you decouple it a bit to show how each component contributes to performance individually ( like INR+some components).

- Can you also your model’s results on datasets such as Cardiac, which are used by baselines like UVI-Net, to better demonstrate generalization and fairness in comparison.

---

> ### Author Response · Authors · 2025-11-25
> **Comment 11**
>
> **Q11: The baseline comparison appears outdated; for example, SVIN may not be the strongest reference point. It would be more convincing to compare against recent natural image super-resolution methods retrained on microscopy data (e.g., [1]).**
>
> Thank you very much for the valuable comments!
> > In SeCo‐INR [1], the authors condition traditional implicit neural representations with local semantic prior information to improve super-resolution for medical images. The core idea is to obtain local segmentation masks through pixel-level segmentation supervision during training. However, this also means that the training process relies on high-quality segmentation results, which limits its applicability in the field of microscope image processing, especially for a vast amount of unlabeled microscope images. SpatimeINR addresses two representative problems in 4D microscope time-series imaging. First, due to the anisotropic resolution inherent in microscope volumetric imaging, there is an insufficient resolution along the z-axis; simultaneously, to prevent issues such as photobleaching caused by excessively high sampling density, the sampling density along the t-axis is limited. Second, live biological samples often exhibit nonlinear and non-periodic dynamic changes over time (**Appendix C, Figure S1**). By designing local spatiotemporal feature encoding and a unified implicit 4D volumetric rendering framework, SpatimeINR provides a super-resolution image enhancement method for 4D microscope imaging that does not depend on paired data. The two methods differ in design philosophy and applicable scenarios. Moreover, since we were unable to find open-source code for SeCo‐INR, we hope to apply for an exemption from comparing with SeCo‐INR.
> >
> > In Sections 5.1 and 5.2 of the main text, we conducted detailed comparisons with DynINR—the latest 4D medical image reconstruction method based on implicit neural representations from 2025 and CuNeRF , a 3D medical volumetric image super-resolution method. Additionally, for the temporal super-resolution task, we introduced the downstream biological metric Biological Shape Accuracy (BSA, **Section 4.3**) to evaluate the super-resolved results. Experimental results indicate that DynINR can achieve good spatiotemporal super-resolution, but it has limitations in capturing complex 4D dynamic spatiotemporal changes. Its temporal super-resolution results exhibit linear changes, which do not accurately reflect the true biological phenomena; whereas CuNeRF, despite possessing certain advantages in spatial continuity, still falls short in generating detailed fine structures.
> >
> > [1] Ekanayake, Mevan, SeCo-INR: Semantically Conditioned Implicit Neural Representations for Improved Medical Image Super-Resolution
>
> We have integrated these discussion into the revised manuscript’s **Section 5.1 (Page 8/Line 419) and 5.2 (Page 9/Line 474)**.

---

> ### Author Response · Authors · 2025-11-25
> **Comment 12**
>
> **Q12: In ablation, instead of testing a combination of three different components, can you decouple it a bit to show how each component contributes to performance individually ( like INR+some components).**
>
> Thank you very much for the valuable comments!
>
> > | INR  | Rendering | CCL  | latent code | PSNR   | BSA    | LPIPS  |
> > |------|-----------|------|-------------|--------|--------|--------|
> > | ✓    |           |      |             | 25.834 | 37.79  | 2.951  |
> > | ✓    | ✓         |      |             | 30.114 | 22.68  | 1.655  |
> > | ✓    | ✓         | ✓    |             | 30.862 | 14.54  | 1.582  |
> > | ✓    | ✓         | ✓    | ✓           | 33.358 | 4.41 | 1.212 |
> >
> > *Table: Quantitative results for different module retention configurations. The leftmost four columns indicate the retention status of the corresponding modules: INR, latent code, Rendering, and CCL. The units of LPIPS and BSA are \(10^{-1}\) and \(10^{-3}\), respectively.*
> >
> > To comprehensively assess the contribution of each module to overall model performance, we conducted ablation studies on the spatiotemporal latent encoding, the 4D rendering, and the cycle consistency loss. The complete model achieved the best performance across evaluation metrics (**Table 3**). The baseline INR resulted in suboptimal reconstruction, exhibiting loss of image details, chessboard artifacts among discrete pixels (**Figure 7**), and the lowest scores across all quantitative measures. Incorporating the 4D rendering significantly improved continuity and precision; however, noticeable grid artifacts and minor noise persisted at the hypercube sampling boundaries. The addition of the cycle consistency module effectively reduced noise levels and maintained brightness uniformity, while providing an indirect constraint for large-scale upsampling in the absence of corresponding ground truth, thus enhancing overall reconstruction fidelity. Nonetheless, the model’s capacity to capture nonlinear motion remained limited, supporting only linear interpolation of intermediate dynamics. With the integration of spatiotemporal latent encoding, the model's ability to perceive the 4D data manifold was substantially enhanced, enabling effective learning of true biological motion trajectories along the temporal axis. For instance, the *SpatimeINR* reconstruction in **Figure 7** accurately reflected the rotational dynamics of the major axis (as indicated by the blue arrow) during human lung cancer cell metastasis. Quantitative evaluations across different latent dimensionalities (**Appendix D**) reveal that the reconstruction quality remains stable, whereas the absence of fundamental motion structures in low-resolution temporal data prevents *SpatimeINR* from recovering accurate super-resolution results (**Appendix E**).
>
> We have integrated these discussion into the revised manuscript’s **Section 5.3 (Page 10/Line 492)**.

---

> ### Author Response · Authors · 2025-11-25
> **Comment 13**
>
> **Q13: Can you also your model’s results on datasets such as Cardiac, which are used by baselines like UVI-Net, to better demonstrate generalization and fairness in comparison.**
>
> Thank you very much for the valuable comments!
>
> > UVI-Net is designed to predict intermediate frames using only the start and end frames. In contrast, SpatimeINR, as an image super-resolution method, requires at least a low-resolution image containing basic structural information to recover authentic details. Without such structural cues, reconstructing subtle variations becomes challenging (**Appendix E**). Consequently, comparing UVI-Net (relies solely on boundary frames) with SpatimeINR (depends on greater temporal information) results in an unequal input information scenario. We designed several comparative experiments.
> > % Table for Cardiac (ACDC) results
> >
> > | Dataset | Metric | UVI-Net | SpatimeINR (inp-2) | SpatimeINR (inp-5) |
> > |---------|--------|---------|---------------------|--------------------|
> > | ACDC    | PSNR   | 33.59   | 28.85               | 34.01              |
> > |         | SSIM   | 0.978   | 0.91                | 0.978              |
> > |         | LPIPS  | 1.066   | 4.428               | 1.012              |
> >
> > *Table: Quantitative results of UVI-Net and _SpatimeINR_ under different input conditions on the ACDC dataset. The unit of LPIPS is 10^-2*
> >
> >
> > On the Cardiac (ACDC) dataset, one scheme provided _SpatimeINR_ only with the first and last frames, while another scheme input five frames (the initial frame, frames at 25%, 50%, and 75% of the sequence, and the final frame) to predict intermediate motion over 10 test samples. UVI-Net was pretrained on 90 datasets and tested on 10 samples in accordance with the strategy described by its authors. Results show that when predicting intermediate motion solely from the boundary frames, _SpatimeINR_ failed to recover the omitted motion information (**Figure S3**), with its PSNR, SSIM, and LPIPS significantly below those of UVI-Net. However, when additional intermediate frames provided a basic motion structure, _SpatimeINR_ markedly improved its reconstruction of the non-periodic cardiac motion and outperformed UVI-Net across all metrics. Given the disparity in input information, these comparisons remain only indicative.
> >
> > % Table for CE results
> >
> > | Dataset | Metric | SpatimeINR | UVI-Net (pred-full) | UVI-Net (pred-interp) |
> > |---------|--------|------------|---------------------|-----------------------|
> > | CE      | PSNR   | 32.4       | 23.33               | 30.44                 |
> > |         | SSIM   | 0.971      | 0.762               | 0.96                  |
> > |         | LPIPS  | 1.25       | 3.946               | 1.73                  |
> >
> > *Table: Quantitative results of _SpatimeINR_ and UVI-Net under different input conditions on the CE dataset.The unit of LPIPS is 10^-1*
> >
> >
> > Furthermore, supplementary comparisons were conducted on the CE dataset, which features complex and irreversible nonlinear variations. In one configuration, UVI-Net employed a self-supervised training protocol consistent with the original paper to predict 28 intermediate frames from the boundary frames only. In another configuration, UVI-Net was pretrained on 12 pairs of boundary data formed through grouping ($\{I_{\mathrm{front}}, I_{\mathrm{interp}}, I_{\mathrm{interp}}, I_{\mathrm{interp}}, I_{\mathrm{end}}\}$) and then predicted 3 intermediate frames per group. Meanwhile, _SpatimeINR_ directly used 5 low-temporal-resolution frames to predict the complete 30-frame high-temporal-resolution sequence. Experimental results indicate that UVI-Net exhibited noticeable prediction errors with only boundary inputs (**Figure S3**); although grouped prediction yielded some improvement, its complexity and heavy preprocessing render it impractical for real microscopic imaging scenarios.
> >
> > In summary, UVI-Net is suitable for predicting motion trajectories from boundary frames in tasks characterized by modest, simple variations (e.g., medical image registration or MRI), whereas _SpatimeINR_ addresses 4D microscopic data with complex dynamics in long sequences, offering a more effective and practical solution.
>
> We have integrated these discussion into the revised manuscript’s **Appendix I (Page 18/Line 918)**.

---

### Official Review · Reviewer_feDB · 2025-10-31

**Soundness:** 2
**Presentation:** 2
**Contribution:** 2
**Rating:** 4
**Confidence:** 3

**Summary:**

The paper proposes SpatimeINR, an implicit neural representation framework for arbitrary-scale spatiotemporal super-resolution of 4D fluorescence microscopy data. The method integrates spatiotemporal latent representations, a conditional MLP, and a 4D volumetric rendering module with cycle-consistency loss. Extensive experiments on lung cancer cell (A549) and C. elegans cell membrane fluorescence datasets demonstrate that SpatimeINR outperforms existing traditional and deep learning methods in reconstructing complex 4D dynamics, both quantitatively and qualitatively.

**Strengths:**

1. The work successfully extends implicit neural representations (INR) to full 4D (3D space + time) biological imaging, enabling continuous and arbitrary-scale reconstruction. This addresses a significant gap in prior work, which often focused on 2D/3D or fixed-scale super-resolution.
2. The method introduces a principled approach to encode per-voxel temporal trajectories as latent codes, effectively capturing dynamic information across both spatial and temporal dimensions (as illustrated in Figure 3).
3. The authors conduct thorough experiments on two challenging real-world datasets, covering both spatial and temporal super-resolution tasks. Ablation studies (Section 5.3) convincingly demonstrate the importance of each component: spatiotemporal encoding, 4D rendering, and cycle-consistency loss.

**Weaknesses:**

1. The use of variable x in Equation (1) is ambiguous—it is used to denote both spatial coordinates and full 4D spatiotemporal coordinates. This should be clarified to avoid confusion.
2. The latent code extraction process (Section 3.2) lacks a clear description of how queries between low-resolution grid points are handled (e.g., nearest-neighbor vs. interpolation). The downsampling operator D(⋅) used in the cycle-consistency loss (Equation 11) is not defined.
3. Several recent and highly relevant INR-based methods for 4D medical imaging are not discussed or compared, including:
Saitta et al. (2024): INR for 4D flow MRI super-resolution.
Xu et al. (2024): Self-supervised INR for 3D fluorescence microscopy.
The omission of these works weakens the positioning of SpatimeINR within the current research landscape.
4. The description of the proposed framework is not comprehensive enough. Critical implementation details for key components—such as the encoder architecture, the latent code sampling strategy, and the feature fusion mechanism in the MLP—are omitted, hindering a complete understanding and reproducibility.
5. Lack inference speed and memory usage.

**Questions:**

The cycle-consistency loss formulation references a downsampling operator $\mathcal{D}(\cdot)$, but its definition is not made explicit. Is this operator simply a uniform average, or does it model specific physical/optical effects? Please clarify its construction and impact on reconstruction fidelity.
Can the authors provide timing benchmarks and memory use for both training and inference across their datasets, and comment on applicability to even larger 4D datasets (for example, developmental time-lapse volumes or multi-field MRI)?

---

> ### Author Response · Authors · 2025-11-25
> **Comment 5**
>
> **Q5: The use of variable x in Equation (1) is ambiguous—it is used to denote both spatial coordinates and full 4D spatiotemporal coordinates. This should be clarified to avoid confusion.**
>
> Thank you very much for the valuable comments!
>
> >We have made a clear distinction between variables in the revised manuscript: we denote the variable representing pure spatial coordinates as *xₛ*, while the complete 4D spatiotemporal coordinates are uniformly denoted as *uₛₜ*, where
> >
> >uₛₜ = (xₛ, t) ∈ [-1, 1]⁴.
> >
> >Consequently, the original Equation (1)
> >
> >Iₕᵣ = { SpatimeINR(τᵢ)(x) }₍ᵢ₌₁₎ᴺ
> >
> >has been revised to
> >
> >Iₕᵣ = { SpatimeINR(τᵢ)(uₛₜ) }₍ᵢ₌₁₎ᴺ,
> >
> >where τᵢ represents the time series signal corresponding to the spatial location *xₛ*.
>
> We have integrated these discussion into the revised manuscript’s **Section 3.1 (Page 4/Line 174)**.

---

> ### Author Response · Authors · 2025-11-25
> **Comment 6**
>
> **Q6: The latent code extraction process (Section 3.2) lacks a clear description of how queries between low-resolution grid points are handled (e.g., nearest-neighbor vs. interpolation).**
>
> Thank you very much for the valuable comments!
>
> >We employ an L1 distance-based nearest neighbor search method for non-integer points during high-resolution sampling. Specifically, for any given input spatial coordinate xₛ ∈ [-1, 1]³, we locate the grid point in the low-resolution feature map that is closest to xₛ in terms of L1 distance; the corresponding latent code z(xₛ) is then extracted as the vector. This lookup process retrieves a latent vector of length d in the 3D space, which integrates the spatial features at the (x, y, z, :) position of the current 4D data (x, y, z, t) along with the temporal features from time 0 to t. Subsequently, we concatenate the query coordinate with the complete d-dimensional vector and input the result into the MLP.
>
> We have integrated these discussion into the revised manuscript’s **Section 3.2 (Page 4/Line 201)**.

---

> ### Author Response · Authors · 2025-11-25
> **Comment 7**
>
> **Q7: Several recent and highly relevant INR-based methods for 4D medical imaging are not discussed or compared, including: Saitta et al. (2024): INR for 4D flow MRI super-resolution. Xu et al. (2024): Self-supervised INR for 3D fluorescence microscopy. The omission of these works weakens the positioning of SpatimeINR within the current research landscape.**
>
> Thank you very much for the valuable comments!
>
> >We supplemented our discussion by highlighting the differences in application scenarios and modeling strategies between SpatimeINR and the 4DflowINR [1] as well as 3DINR [2] methods here. 4DflowINR is based on the characteristics of MRI imaging and physical models of vascular flow velocity. It employs phase-contrast and k-space sampling techniques to comprehensively model the blood flow velocity field and wall shear stress, with its design focused on ensuring the physical consistency of the global blood flow field and satisfying boundary conditions (e.g., no-slip constraints). Regarding the Self-supervised INR for 3D fluorescence microscopy by Xu et al. (2024) that you mentioned, we have only found one work that is relatively similar, though we cannot confirm it with 100% certainty: “Zhou, Y., Xu, C., Jin, Z., Chen, Y., Zheng, B., Wang, M., ... & Gu, N. (2024). Physics-Informed Ellipsoidal Coordinate Encoding Implicit Neural Representation for high-resolution volumetric wide‐field microscopy. bioRxiv, 2024-10.” This method primarily utilizes implicit neural representations to achieve image deblurring and enhance the z-axis resolution, and is suited for static 3D reconstruction of wide-field fluorescence microscopy images, aiming to eliminate the blurring effects caused by the missing cone problem. In contrast, SpatimeINR is designed to address the issues commonly encountered in live microscopic imaging, such as low sampling rates along the t-axis, anisotropic resolution along the z-axis, and nonlinear, dynamic motions within biological samples (especially rapid changes during certain stages of cell development) (**Appendix C**). It builds a continuous modeling approach that combines local spatiotemporal implicit encoding, a conditional multilayer perceptron, and implicit volume rendering, thereby achieving high-precision recovery of microscopic dynamic details at arbitrary scales. Thus, SpatimeINR fundamentally differs from 4DflowINR and 3DINR in terms of design philosophy and application scenarios.
> >
> >It should be noted that the open-source GitHub code for 4DflowINR is currently incomplete (the project description states that it is still under development), and while the related work on 3DINR has been uploaded to bioRxiv, its code has not been released. Therefore, we hope to request an exemption from conducting experimental comparisons for these two methods, and instead discuss only the differences in their design principles and application scenarios in the methodology section. We compared our method with the experiments in Section 5.1 and the latest DynINR method from 2025, and additionally introduced the downstream biological metric Biological Shape Accuracy (BSA, **Section 4.3**) to evaluate the super-resolution results. Experimental results indicate that while DynINR can achieve spatiotemporal super-resolution reasonably well, it has limitations in capturing complex 4D dynamics. Its super-resolution results along the time axis exhibit a linear change, which deviates from the behavior observed in real biological phenomena.
> >
> >[1] Simone Saitta, Implicit neural representations for unsupervised super-resolution and denoising of 4D flow MRI
> >[2] You Zhou, Physics-Informed Ellipsoidal Coordinate Encoding Implicit Neural Representation for high-resolution volumetric wide field microscopy
>
>
> We have integrated these discussion into the revised manuscript’s **Section 5.1 (Page 8/Line 419)**.

---

> ### Author Response · Authors · 2025-11-25
> **Comment 8**
>
> **Q8: The description of the proposed framework is not comprehensive enough. Critical implementation details for key components—such as the encoder architecture, the latent code sampling strategy, and the feature fusion mechanism in the MLP—are omitted, hindering a complete understanding and reproducibility.**
>
> Thank you very much for the valuable comments!
>
> >We employ the RCAN architecture with the following implementation. The input is a 5D tensor of shape $(bsize,X,Y,Z,T)$. After permuting its dimensions to $(bsize,T,X,Y,Z)$, the tensor is fed to a head module composed of a 3D convolution with symmetric padding for preliminary feature extraction. Deep features are then extracted by a main module consisting of multiple ResidualGroup3D blocks. Each ResidualGroup3D contains several RCAB3D modules; within each RCAB3D, two layers of 3D convolution with ReLU activation are applied alongside a CALayer3D—which uses adaptive average pooling and a $1\times1\times1$ convolution to implement channel attention—with the input features combined via a residual connection. In the tail module, a $1\times1\times1$ convolution maps the features to a low-dimensional latent code, and the result is passed through a $\tanh$ function (scaled by $\pi$) for output.
> >
> >The latent code sampling strategy is L1 distance-based nearest neighbor search method for non-integer points during high-resolution sampling. Specifically, for any given input spatial coordinate xₛ ∈ [-1, 1]³, we locate the grid point in the low-resolution feature map that is closest to xₛ in terms of L1 distance; the corresponding latent code z(xₛ) is then extracted as the vector. This lookup process retrieves a latent vector of length d in the 3D space, which integrates the spatial features at the (x, y, z, :) position of the current 4D data (x, y, z, t) along with the temporal features from time 0 to t. Subsequently, we concatenate the query coordinate with the complete d-dimensional vector and input the result into the MLP.
> >
> >For the conditional MLP, our revised manuscript details its implementation as follows. The MLP encodes input coordinates at multiple scales using a preset maximum frequency along with sine and cosine functions (**Equation 6**), yielding an embedding vector. This encoding is concatenated with additional channel information (the sampling results from **Section 3.2**) to form the complete input, which is then processed by a network of fully connected layers. The MLP comprises eight hidden layers, each with 256 neurons and ReLU activation. In the fourth layer, a predefined skip connection concatenates the complete input with the current output to reinforce the recovery of high-frequency details, this is the feature fusion mechanism. Finally, a linear layer produces the target output dimension.
>
> We have integrated these discussion into the revised manuscript’s **Appendix F (Page 16/Line 844)**.

---

> ### Author Response · Authors · 2025-11-25
> **Comment 9**
>
> **Q9: The downsampling operator D(⋅) used in the cycle-consistency loss (Equation 11) is not defined.；The cycle-consistency loss formulation references a downsampling operator , but its definition is not made explicit. Is this operator simply a uniform average, or does it model specific physical/optical effects? Please clarify its construction and impact on reconstruction fidelity.**
>
> Thank you very much for the valuable comments!
>
> >In this paper, we employ a simple average downsampling method to construct the operator. Specifically, for a high-resolution image denoted as I_hr, we partition it into several regions W and define the downsampling operator D(⋅) as follows:
> >
> >$$
> >D(I_{hr})(x, y, z, t) = \frac{1}{|W|} \sum_{i \in W} I_{hr}(i)
> >$$
> >
> >where |W| represents the number of pixels (or voxels) in region W. The use of average downsampling is primarily motivated by the following considerations:
> >
> >1. Average downsampling can effectively simulate the continuity of dynamic changes along the *t*-axis in fluorescence microscopy images, ensuring consistency in both noise levels and overall brightness in the reconstructed image.
> >2. For large-scale upsampling results that lack corresponding large-scale ground truth, direct fitting may lead to issues such as holes, speckles, or local brightness discontinuities. In contrast, average downsampling provides an indirect constraint that helps enhance the overall reconstruction fidelity.
> >
> >Furthermore, we quantify the effect of the cycle consistency loss through ablation experiments (**Section 5.3**). The introduction of the cycle consistency module not only effectively mitigates inconsistencies in noise levels and overall brightness but also provides an indirect constraint for large-scale upsampling results without corresponding ground truth. This approach helps eliminate grid-like artifacts (**Figure 7**) and contributes to improved overall reconstruction fidelity (**Table 3**).
>
> We have integrated these discussion into the revised manuscript’s **Section 3.4 (Page 5/Line 263) and 5.3 (Page 10/Line 492)**.

---

> ### Author Response · Authors · 2025-11-25
> **Comment 10**
>
> **Q10: Lack inference speed and memory usage.；Can the authors provide timing benchmarks and memory use for both training and inference across their datasets, and comment on applicability to even larger 4D datasets (for example, developmental time-lapse volumes or multi-field MRI)?**
>
> Thank you very much for the valuable comments!
>
> >
> > | Dataset       | Train               | Inference (×2)    | Inference (×4)    | GPU Memory |
> > |---------------|---------------------|-------------------|-------------------|------------|
> > | CE (t=15)     | 22                  | 18                | 35                | 4000       |
> > | A549 (t=15)   | 21                  | 16                | 33                | 4000       |
> > | ACDC (t=19)   | 21                  | 20                | 35                | 5000       |
> > | Syn (t=240)   | CUDA out of memory  |                   |                   |            |
> > | Syn* (t=240)  | 55                  | 22                | 39                | 9000       |
> >
> > *Table: Computation report of SpatimeINR on diverse datasets, including two synthetic ultra-large datasets.*
> >
> >
> > We systematically evaluated five four-dimensional datasets on an NVIDIA RTX 4090. The CE dataset has volume dimensions of $(15,128,178,80)$ (ordered as $t,x,y,z$), the A549 dataset is $(15,150,175,115)$, and the ACDC (cardiac MRI) dataset is $(19,180,224,10)$. Additionally, by resizing eight groups (each comprising 30 time points) of the CE and A549 datasets to match the CE dimensions and concatenating them along the temporal axis, we constructed a large-scale dataset with volume dimensions of $(240,128,178,80)$ for evaluating *SpatimeINR* on extensive data.
> >
> > Experimental results indicate that training and inference at 2× scale require approximately 20 minutes per dataset, while under 4× super-resolution the inference time stays below 40 minutes, with an almost linear increase in processing time. After optimizing with `torch.cuda.empty_cache()`, the peak GPU memory usage remains under 5000 MB, averaging around 4000 MB, which demonstrates that the method runs efficiently on most GPUs. For the synthetic dataset Syn, the original data size caused CUDA memory shortages. By down-sampling the $x$, $y$, and $z$ axes by a factor of two while preserving temporal resolution, we obtained a dataset, denoted Syn*, with volume dimensions of $(240,64,89,40)$; its training time is approximately 55 minutes, and both inference time and memory consumption are within acceptable limits. These results suggest a practical strategy for processing larger four-dimensional data by reducing volume resolution—with care to preserve critical information—to lower training time and memory load.
>
> We have integrated these discussion into the revised manuscript’s **Appendix H (Page 17/Line 888)**.

---

### Official Review · Reviewer_6Q1x · 2025-11-02

**Soundness:** 3
**Presentation:** 3
**Contribution:** 3
**Rating:** 6
**Confidence:** 2

**Summary:**

The paper introduces an implicit neural representation framework for arbitrary-scale 4D spatiotemporal super-resolution in fluorescence microscopy. By modeling volumetric time-varying data as a continuous implicit function and combining local sampling, a conditional MLP, and cycle-consistency constraints, the method reconstructs high-resolution 4D sequences from low-resolution inputs without relying solely on boundary frames. Experiments on lung cancer cell and *C. elegans* microscopy datasets show that SpatimeINR outperforms both traditional interpolation and deep learning baselines, preserving fine structures and improving downstream segmentation performance, demonstrating strong capability in handling complex, non-periodic cellular dynamics.

**Strengths:**

- The paper makes a meaningful contribution by extending arbitrary-scale implicit neural representations to continuous 4D spatiotemporal microscopy, a setting rarely addressed in prior SR research. The combination of spatiotemporal latent codes, local INR rendering, and cycle-consistent degradation represents a thoughtful hybrid of existing ideas adapted to biological imaging constraints.
- The technical components are well-motivated and carefully implemented: latent trajectory encoding per voxel, mixed uniform and importance sampling for 4D coordinates, and conditional MLP rendering. The experiments are rigorous, covering multiple scaling factors, two biological datasets, and downstream segmentation with MedSAM. Ablations clearly verify the necessity of each module.
- The paper is clearly written with intuitive figures and consistent mathematical notation. The model architecture, training losses, and data preprocessing steps are explained in detail, enabling reproducibility.

**Weaknesses:**

Need more ablation studies to quantify the sensitivity of the model to architectural and data-related design choices, including:

- the effect of latent dimensionality on spatiotemporal fidelity,

- the role of temporal context length during encoding,

- robustness to varying noise levels or photobleaching patterns.

**Questions:**

- How does SpatimeINR fundamentally differ from dynamic NeRF / time-conditioned neural fields (e.g., D-NeRF, Nerfies, HyperNeRF) or grid-based spatiotemporal implicit models? Clarifying this distinction and ideally including at least one representative comparison (or explaining why not) would help position the contribution

- Why adopt a voxel-wise latent trajectory encoding instead of alternatives like continuous temporal embeddings or learned spatiotemporal hash grids? Empirical or conceptual justification would strengthen the design motivation.

- Could the authors provide qualitative examples where SpatimeINR fails or struggles (e.g., extremely rapid motion, highly anisotropic data, low SNR situations)?

---

> ### Author Response · Authors · 2025-11-25
> **Comment 1**
>
> **Q1: the effect of latent dimensionality on spatiotemporal fidelity,**
>
> Thank you very much for the valuable comments!
>
> >| Latent Dim       | 0      | 32     | 64     | 128    | 256    |
> >| ---------------- | ------ | ------ | ------ | ------ | ------ |
> >| **PSNR**       | 30.862 | 31.654 | 32.286 | 33.358 | 33.251 |
> >| **BSA (×10⁻³)** | 14.54  | 6.50   | 4.93   | 4.708  | 5.03   |
> >| **Time (min)**   | 16     | 20     | 20     | 22     | 27     |
> >
> >We investigated the impact of the latent dimension on spatiotemporal fidelity by varying only the latent code dimension while keeping the remainder of the network unchanged. The latent dimension was set to 0, 32, 64, 128, and 256, and reconstruction performance on the CE dataset at 4× scale was evaluated using PSNR, Biological Shape Accuracy (BSA, **Section 4.3 benchmark**), and training time. Results show that without latent code (dimension 0), the model required 16 minutes to train but yielded substantially worst PSNR and BSA. At a dimension of 32, both PSNR and BSA are better, with training time increasing to 20 minutes. When the dimension ranged from 64 to 256, the PSNR and BSA metrics remained stable, with training time reaching 27 minutes at 256 dimensions. Although the training time increased progressively with the latent dimension, its benefit to reconstruction quality was marginal.
>
> We have integrated these discussion into the revised manuscript’s **Appendix D (Page 15/Line 768)**.

---

> ### Author Response · Authors · 2025-11-25
> **Comment 2**
>
> **Q2: the role of temporal context length during encoding,；Could the authors provide qualitative examples where SpatimeINR fails or struggles (e.g., extremely rapid motion, highly anisotropic data, low SNR situations)?；robustness to varying noise levels or photobleaching patterns.**
>
> Thank you very much for the valuable comments!
>
> >| Temporal Context Length | 30     | 15     | 10     | 5      | 2      | 1      |
> >| ----------------------- | ------ | ------ | ------ | ------ | ------ | ------ |
> >| **PSNR**              | 36.613 | 34.12  | 32.562 | 31.204 | 23.41  | 19.88  |
> >| **BSA (×10⁻³)**        | 2.623  | 3.125  | 3.99   | 4.09   | 64.387 | 97.756 |
> >
> >
> >| Noise Level (σ)       | 0      | 5      | 10     | 30     | 50     | 100    |
> >| --------------------- | ------ | ------ | ------ | ------ | ------ | ------ |
> >| **PSNR**            | 33.764 | 33.358 | 32.84  | 29.939 | 27.83  | 25.522 |
> >| **BSA (×10⁻³)**       | 3.571  | 3.581  | 6.796  | 33.821 | 47.266 | 78.619 |
> >
> >We supplemented our study with a systematic investigation of the effect of temporal context length during encoding. *SpatimeINR* was trained with input sequences of length $t=30$ (SR factor 1), $t=15$ (2×), $t=10$ (3×), $t=5$ (6×), $t=2$ (15×), and $t=1$ (30×), and then used to predict the complete temporal sequence at $t=30$. Experimental results indicate that when $t \geq 5$, the overall performance remains stable despite slight declines in prediction accuracy. In contrast, at $t=2$ or $t=1$, the input data loses its basic motion structure on the temporal axis, leading to a marked failure in super-resolution reconstruction. This behavior is analogous to a microscope being unable to capture a complete motion process under very rapid motion. To further simulate scenarios with severe anisotropy, we conducted a 10-fold downsampling experiment along the $z$-axis. The results reveal a significant deterioration in reconstruction quality (**Appendix E**, **Figure S2ab**).
> >
> >Concurrently, we evaluated model robustness by adding Gaussian noise with $\sigma=0$ (noise-free), 5 (mild), 10 (mild), 30 (moderate), 50 (severe), and 100 (severe) to the images. The experiments show that *SpatimeINR* is robust to mild noise, with negligible changes in performance; however, under moderate or higher noise levels, both PSNR and BSA deteriorate considerably, and the preservation of fine details in downstream segmentation tasks is greatly diminished (**Appendix E**, **Figure S2b**). These findings suggest that moderate denoising is advisable in practical applications.
>
> We have integrated these discussion into the revised manuscript’s **Appendix E (Page 15/Line 790)**.

---

> ### Author Response · Authors · 2025-11-25
> **Comment 3**
>
> **Q3: How does SpatimeINR fundamentally differ from dynamic NeRF / time-conditioned neural fields (e.g., D-NeRF, Nerfies, HyperNeRF) or grid-based spatiotemporal implicit models? Clarifying this distinction and ideally including at least one representative comparison (or explaining why not) would help position the contribution**
>
> Thank you very much for the valuable comments!
>
> >Dynamic NeRF methods (such as D-NeRF, Nerfies, HyperNeRF) primarily target multi-view synthesis tasks in dynamic scenes, whereas our work focuses on the issues of insufficient resolution along the z-axis and sparse sampling along the t-axis in 4D microscopic imaging, as well as the non-linear, aperiodic dynamic variations present in biological data (**Appendix C**, **Figure S1**). From a task perspective, arbitrary-scale spatiotemporal super-resolution and multi-view synthesis show significant differences in both requirements and implementations. Without extensive modifications to their code, traditional dynamic NeRF methods cannot be directly applied to spatiotemporal super-resolution tasks.
> >
> >Secondly, in terms of model architecture, dynamic NeRF methods generally adopt an INR-only structure, which performs very well in the multi-view synthesis domain. In contrast, current arbitrary-scale super-resolution methods (such as [1,2,3], and recent Gaussian splatting super-resolution methods like [4,5]) typically employ an Encoder-INR architecture, where the encoder is used to extract features from low-resolution images and a conditional MLP utilizes these features in combination with arbitrary-scale coordinates to render super-resolved images. In Sections 5.1 and 5.2 of the main text, we provide detailed comparisons in both the temporal and spatial dimensions with INR/NeRF methods (such as DynINR [6] and CuNeRF [7]). The results indicate that DynINR shows limitations when capturing the complex 4D spatiotemporal dynamics; its super-resolution performance along the time axis exhibits linear changes that fail to reflect real biological phenomena. Meanwhile, although CuNeRF demonstrates certain advantages in continuous spatial representation, it is unable to generate sufficiently detailed structural features. Moreover, our supplementary ablation experiments (**Section 5.3**) further prove that removing the encoder module leads to a marked deterioration in the reconstruction of temporal dynamics and 3D structures.
> >
> >In summary, we contend that the SpatimeINR super-resolution method is fundamentally different from multi-view synthesis-based dynamic NeRF methods, and our comparisons with INR-only methods in the main text emphasize the advantages of our approach.
> >
> >
> >[1] Y.Chen, Learning continuous image representation with local implicit image function.
> >[2] Chen, Z  Videoinr: Learning video implicit neural representation for continuous space-time super-resolution.
> >[3] Xie, Q Rotation Equivariant Arbitrary-scale Image Super-Resolution.
> >[4] Chen, D, Generalized and Efficient 2D Gaussian Splatting for Arbitrary-scale Super-Resolution.
> >[5] Peng, L., Pixel to gaussian: Ultra-fast continuous super-resolution with 2d gaussian modeling.
> >[6] Jie Feng, Spatiotemporal Implicit Neural Representation for Unsupervised Dynamic MRI Reconstruction
> >[7] Zixuan Chen, CuNeRF: Cube-Based Neural Radiance Field for Zero-Shot Medical Image Arbitrary-Scale Super Resolution
>
>
>
>
> We have integrated these discussion into the revised manuscript’s **Section 2.1 (Page 2/Line 103)**.

---

> ### Author Response · Authors · 2025-11-25
> **Comment 4**
>
> **Q4: Why adopt a voxel-wise latent trajectory encoding instead of alternatives like continuous temporal embeddings or learned spatiotemporal hash grids? Empirical or conceptual justification would strengthen the design motivation.**
>
> Thank you very much for the valuable comments!
>
> >As shown in **Equation (6)** of the paper, we indeed performed a continuous temporal embedding on the input x, y, z, t coordinates, mapping the raw coordinates into a high-dimensional feature space. However, relying solely on coordinate embeddings makes it difficult to capture the rich local structure and semantic information present in low-resolution images, which results in INR-only structures being able to perform only simple, smooth predictions based solely on the coordinates in super-resolution tasks. To address this issue, we adopted a voxel-wise latent trajectory encoding method, which extracts contextual priors from low-resolution images via an Encoder, thereby providing the implicit function with the necessary basis to recover high-frequency details. We discuss the reliability of voxel-level latent trajectory encoding in detail in Replay 3, and in Section 5.1 of the main text we compared it with the DynINR method that incorporates hash grid encoding. The experiments demonstrate that although DynINR can generate continuous 4D images, its representation along the temporal axis exhibits linear changes that fail to faithfully capture the nonlinear dynamics in biological data, further proving the superiority of our method in super-resolution tasks. Our supplementary ablation experiments (Section 5.3) further prove that removing the encoder module leads to a marked deterioration in the reconstruction of temporal dynamics and 3D structures.
>
> We have integrated these discussion into the revised manuscript’s **Section 5.3 (Page 7/Line 338)**.

---

### Author Response · Authors · 2025-11-29
**Cover Letter**

**November 30, 2025**

**Dear ICLR Area Chairs,**

We would like to submit our paper “**Implicit Reconstruct Spatiotemporal Super-Resolution Microscopy in Arbitrary Dimension**” for your consideration as an Article in **ICLR2026**.

>In 4D spatiotemporal microscopy, volumetric images of biological samples are acquired by sequentially scanning the z-axis in 3D space. In addition, repeated scans at fixed time intervals on the same living sample yield 3D+t data that incorporate the temporal dimension. This imaging modality is critical for documenting the complex dynamic changes in cells and tissues during biological processes such as embryonic development, cancer cell invasion and metastasis, and neuronal network reconstruction.
>
>However, conventional z-axis slicing methods inherently suffer from anisotropic resolution. Furthermore, to prevent excessive irradiation-induced photobleaching, temporal imaging is typically constrained by limited scanning frequency, resulting in insufficient temporal resolution. These limitations may lead to the loss of fine details in volumetric reconstructions and the omission of key transient events. Consequently, both the precision of dynamic monitoring and the reliability of subsequent quantitative analyses in 4D spatiotemporal microscopy are compromised. Therefore, it is imperative to design an algorithm that simultaneously achieves super-resolution reconstruction along both the spatial and temporal dimensions, while accurately capturing cellular motion.
>
>To address these challenges, we propose a precise and robust spatiotemporal super-resolution method—**SpatimeINR**. Based on an implicit neural representation framework, our approach integrates spatiotemporal coordinates and latent code as inputs to achieve super-resolution reconstruction of 4D data at arbitrary scales along both the spatial and temporal dimensions. We evaluated the proposed method on two 4D datasets: one capturing cell division during embryogenesis in *Caenorhabditis elegans* and another recording amoeboid motion during human lung cancer cell metastasis. Quantitative assessments of the reconstruction quality were conducted, followed by segmentation of the super-resolved images to evaluate the accuracy in recapitulating cellular motion morphology. Experimental results demonstrate that **SpatimeINR** significantly outperforms current methods with respect to both image reconstruction quality and the fidelity of recovered cellular motion. Moreover, after a single training session, the method can concurrently achieve arbitrary-scale super-resolution along both the spatial and temporal dimensions, while offering markedly enhanced user-friendliness compared to existing methods. This method provides biologists with a tool for precise quantitative analysis and dynamic monitoring under microscopy, thereby advancing research into cellular motion, morphological changes, and intercellular interactions, and demonstrating clear practical application scenarios.
>
>We believe that our paper falls within the scope of **ICLR’s applications to physical sciences (physics, chemistry, biology, etc.)** and will attract interest from researchers in microscopy imaging, computer vision, biophysics, and developmental biology.

We appreciate your time and efforts in handling our paper.

**Yours sincerely,**

Anonymous Authors
Department of Biology
Department of Electrical Engineering
Department of Computer Science

---

> ### Author Response · Authors · 2025-12-02
> **Rebuttal Summary**
>
> We appreciate the reviewers' positive and constructive comments on the manuscript!
>
> In the first round of reviews, the reviewers acknowledged the value of our work in terms of its research motivation and its application of implicit neural representations to continuous 4D spatiotemporal super-resolution microscopy reconstruction. The experimental results demonstrate the method’s advantage when applied to challenging real-world datasets, and the paper is well-written and easy to follow. However, several weaknesses were also noted. We summarize the reviewers' comments, along with our responses and revisions, as follows:
>
> **General Comment 1: Addition of Ablation Study and Sensitivity Analysis (Reviewer 6Q1x, Reviewer yv2D)**
> >**Response:** We have added sensitivity analyses in Comments 1 and 2, and redesigned the ablation study in Comment 12 to clearly demonstrate the actual contributions of each module.
>
> **General Comment 2: Insufficient Explanation of Method Details (Reviewer feDB, Reviewer 4zRG)**
> >**Response:** Detailed explanations and supplementary clarifications addressing the method details raised by the reviewers have been provided in Comments 5, 6, 8, 9, and 15.
>
> **General Comment 3: More Comparison and Discussion with Related Methods (Reviewer 6Q1x, Reviewer feDB, Reviewer 4zRG, Reviewer yv2D)**
> >**Response:** We have conducted thorough discussions and comparisons concerning the methods and datasets mentioned by the reviewers in Comments 3, 7, 11, 13, and 16, further clarifying the application scenarios for SpatimeINR.
>
> **General Comment 4: Model Practicality and Performance Metrics (Reviewers feDB, Reviewer 4zRG)**
> >**Response:** We have provided a detailed discussion on the computational report and the reusability of features in Comments 10 and 15.
>
> **General Comment 5: Presentation of Failure Cases and Limitations (Reviewer 6Q1x)**
> >**Response:** In Comment 2, we have presented images of failure cases along with quantitative analyses and a discussion of the limitations of SpatimeINR.
>
> **General Comment 6: Further Elaboration on Novelty (Reviewers yv2D, Reviewer 4zRG)**
> >**Response:** We have supplemented the discussion on the motivation, design rationale, and application scenarios of the SpatimeINR method in Comment 14.
>
> We answered all major and minor comments raised by the four reviewers through additional experiments, literature review, and further clarifications. Accordingly, we revised the Introduction, Related Work, Methods, Quantitative Comparison, and Ablation Study sections in the revised manuscript, and integrated the supplementary experimental results into the Appendix to enhance the overall completeness of the paper.
>
> Although technical issues prevented us from receiving direct feedback from the reviewers during the rebuttal period, we sincerely hope that our rebuttal comments and revised manuscript can thoroughly address the weaknesses and questions raised.

---

### Note · Authors · 2026-01-26

I have read and agree with the venue's withdrawal policy on behalf of myself and my co-authors.

---

### Meta-Review · Area_Chair_2y2g · 2025-12-11

**Summary:**

The paper addresses an important microscopy application, namely the reconstruction of time-varying biological structure from fluorescnece microscopy, and the empirical results are reasonably strong. However, the method is largely an incremental combination of established INR and SR components, with limited conceptual novelty and no biologically informed modeling. Reviewers also raised concerns about the positioning of the contribution relative to recent work in related domains, and were not satisfied with the baseline comparisons. Only some of this could, in my view, be addressed in the rebuttal (e.g. it is understandable to not compare with baselines that do not provide code). Moreover, several technical details are insufficiently specified in the main paper. Given these issues, the submission does not meet the ICLR acceptance bar.

**Reviewer Concerns:**

See meta review above

**Reviewer Scores:**

See meta review

---

### Decision · Program_Chairs · 2026-01-26

Reject